# Signatures of mitonuclear coevolution in a warbler species complex

Silu Wang [1,5 ✉], Madelyn J. Ore[1,6], Else K. Mikkelsen[1,7], Julie Lee-Yaw[2,8], David P. L. Toews [3], Sievert Rohwer[4] & Darren Irwin [1]

Divergent mitonuclear coadaptation could facilitate speciation. We investigate this possibility in two hybridizing species of warblers, *Setophaga occidentalis* and *S. townsendi*, in western North America. Inland *S. townsendi* harbor distinct mitochondrial DNA haplotypes from those of *S. occidentalis*. These populations also differ in several nuclear DNA regions. Coastal *S. townsendi* demonstrate mixed mitonuclear ancestry from *S. occidentalis* and inland *S. townsendi*. Of the few highly-differentiated chromosomal regions between inland *S. townsendi* and *S. occidentalis*, a 1.2 Mb gene block on chromosome 5 is also differentiated between coastal and inland *S. townsendi*. Genes in this block are associated with fatty acid oxidation and energy-related signaling transduction, thus linked to mitochondrial functions. Genetic variation within this candidate gene block covaries with mitochondrial DNA and shows signatures of divergent selection. Spatial variation in mitonuclear ancestries is correlated with climatic conditions. Together, these observations suggest divergent mitonuclear coadaptation underpins cryptic differentiation in this species complex.

[1] Department of Zoology, and Biodiversity Research Centre, 6270 University Blvd, University of British Columbia, Vancouver, BC, Canada. [2] Department of Botany, 3200-6270 University Blvd, University of British Columbia, Vancouver, BC, Canada. [3] Department of Biology, The Pennsylvania State University, University Park, PA, USA. [4] Department of Biology and Burke Museum, Box 353010, University of Washington, Seattle, WA, USA. [5] Present address: Department of Integrative Biology, University of California, Berkeley, CA, USA. [6] Present address: Cornell Lab of Ornithology, Ithaca, NY, USA. [7] Present address: Department of Ecology and Evolutionary Biology, University of Toronto, Toronto, ON, Canada. [8] Present address: Biological Sciences, 4401 University Drive, University of Lethbridge, Lethbridge, AB, Canada. ✉email: siluwang.evo@gmail.com

Mitochondrial (mtDNA) and nuclear (nDNA) genomes interact synergistically in maintaining critical functions that influence fitness in nearly all eukaryotes[1–5]. Populations in different areas may harbour distinct mtDNA sequences due to selection or drift. The mitochondrial genome and its co-functioning nuclear genes are expected to co-evolve, each being the target of selection favoring compatibility[4,6,7]. Hybridization can lead to sub-optimal mitonuclear combinations that could be selected against[8,9]. Therefore, mitonuclear incompatibilities can facilitate divergence between species/populations[10,11]. Hence divergent mitonuclear coevolution is increasingly recognized as being important to speciation[10–12].

Pleistocene glacial expansion and recession have influenced present geographic distribution of mtDNA and nDNA within and between species[13–15]. Mitochondrial DNA has often been viewed as a neutral marker, such that ecological selection and interactions with the nuclear genome have rarely been investigated in natural populations. In at least some species, geographic variation in climate is known to select for different mitochondrial genotypes in different areas[6,16–18]. This may in turn lead to indirect selection on co-functioning nuclear genes. Such climatic mitonuclear coadaptation can lead to genomic differentiation between populations inhabiting different climatic conditions[4,6]. Here we examine the relationship between mtDNA and nDNA variation in a warbler species complex with hybridization. In particular, we ask whether there is a signature of mitonuclear coevolution and, if so, whether climate-related divergent selection could have driven such coevolution and resulted in cryptic divergence among populations experiencing divergent climatic conditions.

When populations under divergent selection come into secondary contact, the result can be an admixed population over a broad region[19–21]. Such populations could become differentiated from the parental populations and even become independent evolutionary trajectories, i.e., hybrid species[22]. Such admixed populations that are semi-isolated from parental populations have the potential to reveal strong selection against suboptimal combinations of genes from the two parental species. Despite increasing interest in mitonuclear interactions at species boundaries of natural populations with complex population histories[3,6,23–26], the degree to which mitonuclear interactions are important in the differentiation among lineages is not well understood.

The North American warbler species *Setophaga occidentalis* (abbreviated as SOCC) inhabits coastal conifer forests along the states of Oregon, California, and southern Washington, USA. To the north of SOCC, a closely related species *Setophaga townsendi* (abbreviated as STOW) consists of an inland population (abbreviated as inland STOW) that inhabits areas east of the Coast Mountains of British Columbia, Canada, and northern Washington, USA, and a coastal population (abbreviated as coastal STOW) west of the Coast Mountains (Fig. 1a)[13,27–30]. The SOCC and inland STOW populations demonstrate distinct plumage and mtDNA haplotypes that are separated by ~0.8% sequence divergence, leading to a divergence time estimate of about 0.5 million years ago (Fig. 1b, c)[13,28,29,31]. The nDNA differences between SOCC and inland STOW are restricted in a few small genomic regions, whereas the rest of the genome shows very little differentiation[31].

Coastal STOW tend to be identical to inland STOW in plumage, but harbor both SOCC and inland STOW mtDNA haplotypes (Fig. 1b, c). The haplotype network (Fig. 1c) suggests that this population is the product of hybridization between SOCC and inland STOW[29] (Fig. 1a). If so, individuals sampled from coastal STOW should demonstrate a mixture of ancestry between SOCC and inland STOW, especially detectable in genomic regions that are differentiated between SOCC and inland STOW.

In addition, the nuclear encoded genes with mitochondrial functions might coevolve with the SOCC versus inland STOW mitochondrial genomes, and be differentiated between coastal and inland STOW.

Here, we examine the role of mitonuclear coevolution in genomic differentiation between hybrid and parental populations. First, we test whether both nuclear and mitochondrial genomic data are consistent with hybrid origin of coastal STOW. Then we examine whether the mtDNA divergence results in amino acid change in protein coding genes, which could cause mitonuclear coevolution. If so, the coevolution is expected to lead to covariance between mtDNA and its co-functioning nuclear genes within and among sites, as individuals with mismatched mt-nDNA genotypes are selected against. Such a force could be counteracted by random mating which breaks down the mt-nDNA association; thus strong selection is required to maintain adaptive mt-nDNA combinations within a single randomly mating population. Over time, however, specific geographic regions may favor particular mtDNA variants and thus the compatible nDNA variants, increasing mt-nDNA concordance among sampling sites. We further investigated signatures of divergent selection on mitonuclear coevolving genes and whether such selection is underpinned by climatic divergence among habitats of hybrids and parental populations. If so, spatial variation in mitonuclear ancestry should covary with climatic variation.

We confirm the hybrid origin of the coastal STOW populations, which are differentiated from the inland STOW in a few genomic regions. One such region of differentiation on chr5 is also highly differentiated between SOCC and inland STOW. This gene block harbors genes that are directly associated with mitochondrial functions, coevolves with mtDNA, and demonstrates signatures of climate-related divergent selection. These results provide evidence that mitonuclear coevolution underpins cryptic differentiation between hybrid and parental lineages in this species complex in the early stage of divergence.

## Results

**Mitochondrial genomic divergence.** Consistent with earlier results based on a small portion of the mtDNA[29], full mitochondrial genomes are distinct between inland STOW and SOCC, with ~0.8% mtDNA sequence divergence between the two clades (Fig. 1c, Table S1). Among the nucleotide substitutions between SOCC and inland STOW, there are 5 amino acid changes: 4 within *ND2* and 1 within *ATP6* (Table S1). Various coastal STOW sampling sites contain a mixture of inland STOW haplotypes and SOCC haplotypes (Fig. 1c), suggesting that these coastal STOW populations are hybrid populations between inland STOW and SOCC[29].

Of the five amino acid substitutions, two (N150D and P320S of *ND2*) were identified as likely to be non-neutral by Provean[32,33] (scores of −3.964 and −2.888, respectively), and one (N150D of *ND2*) was identified as putatively evolving under positive selection for the physicochemical property "Alpha-helical Tendencies" (category 6, $Z = 5.262$) (Table S2; Supplementary Data 3). MEME[34] identified site 25 of *ATP6* as putatively undergoing episodic diversifying selection in Parulidae ($p < 0.0001$), with inland STOW receiving the highest Empirical Bayes Factor among all taxa in the alignment ($EBF = 1 \times 10^{26}$) (Table S2). Together, these results suggest functional differences between the inland STOW and SOCC mitochondrial haplotypes.

**Nuclear genomic differentiation.** Nuclear genomic variation as assessed through variation in 222,559 single nucleotide polymorphisms (SNPs) also reveals clear differentiation between

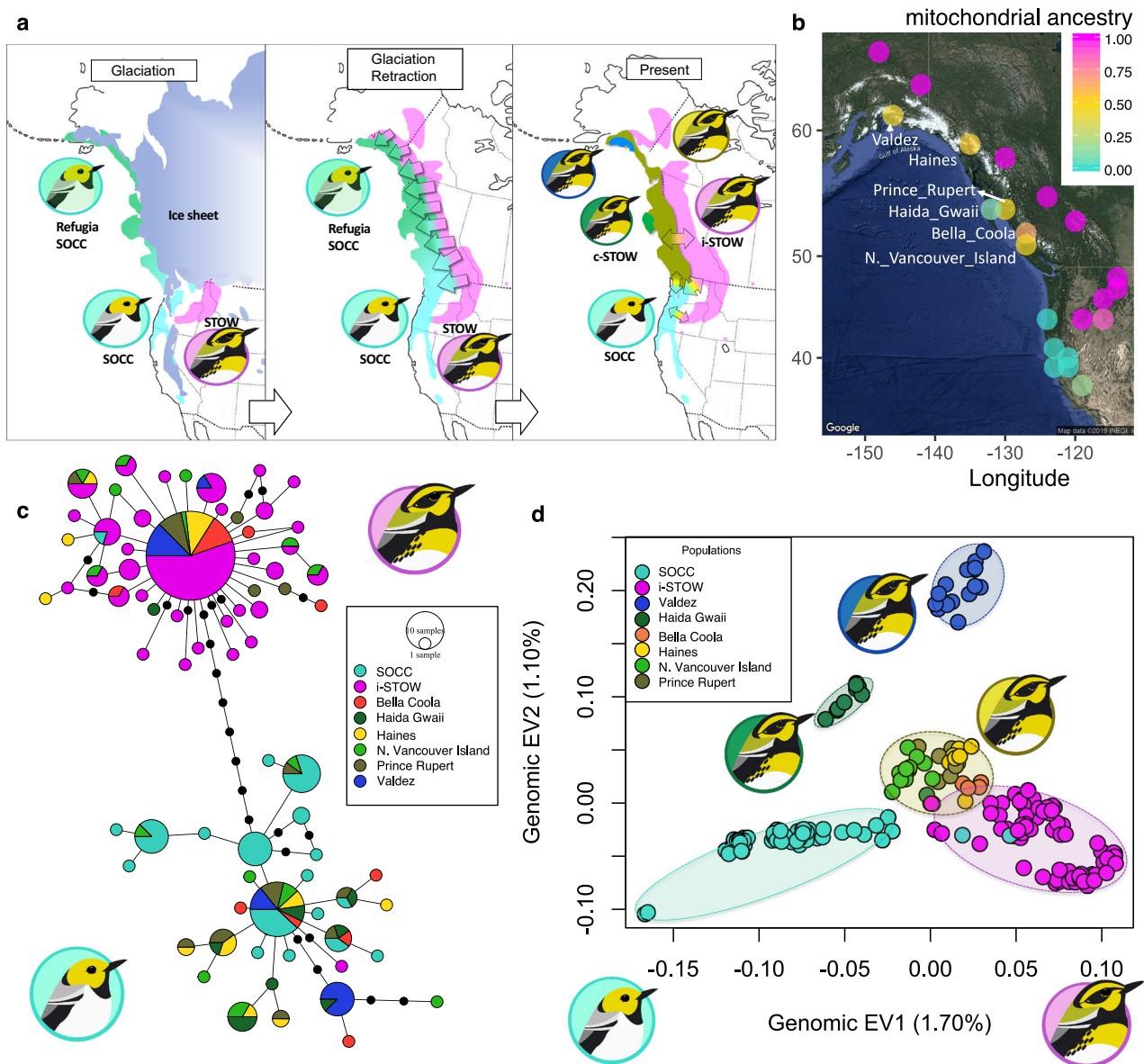

**Fig. 1 Coastal populations of S. townsendi (coastal STOW) demonstrate a mixture of mitochondrial and nuclear ancestry from S. occidentalis (SOCC) and inland S. townsendi (i-STOW). a** Illustration of the inferred history of differentiation and hybridization between STOW and SOCC during glacial expansion and retraction[29]. Left: During the last glacial maxima, the SOCC and STOW populations resided in isolated glacial refugia. Center: after glacial retraction, the refugial SOCC and inland STOW expanded and hybridized along a broad inland-to-coastal front parallel to the coast. Right: the historical hybridization resulted in coastal STOW populations with admixed ancestry although the plumage resembles that of inland STOW. Population substructure within coastal STOW could be a result of refugial isolation. At present, most of the coastal STOW are not isolated from inland STOW by physical barriers. Map data ©2019 Google. Glacial reconstruction illustrations by Silu Wang. **b** Map data ©2019 Google. Mitochondrial ancestry of across sampling site, with colour representing the proportion of individuals belonging to the two major clades shown in: **c** Haplotype network of mitochondrial ND2 gene sequences from ref. [29], with colours representing sampling sites[29] (red for Bella Coola, forest green for Haida Gwaii, yellow for Haines, light green for North Vancouver Island, army brown for Prince Rupert, royal blue for Valdez). Each circle represents a haplotype and area of the circles are proportional to the number of individuals carrying each haplotype. The lines (regardless of their lengths) between the circles represent one mutation between haplotypes, the black dots on the lines represent additional mutations among haplotypes. SOCC and inland STOW almost universally belong to divergent mtDNA clades, whereas coastal STOW populations harbor a mixture of these two major clades. **d** Principle component analysis of covariance with 222,559 SNPs in the nuclear genome. Coastal STOW are intermediate in Eigenvector (EV) 1 but distinct from inland STOW and SOCC in EV2. EV1 represents inland STOW vs SOCC differentiation. Three distinct coastal STOW clusters are evident: Valdez (royal blue), Haida Gwaii (forest green), and other coastal STOW (thereafter abbreviated as oc-STOW, army brown circling gold). Bird illustrations by Gil Jorge Barros Henriques.

SOCC and inland STOW, with coastal STOW forming several additional clusters in the principal components analysis (PCA; Fig. 1d). Inland STOW and SOCC form mostly non-overlapping clusters differing in the first eigenvector (EV1) (Fig. 1d), whereas most coastal STOW individuals have a somewhat intermediate

position. Coastal STOW has substantial substructure, with Valdez and Haida Gwaii forming distinct genetic clusters (Fig. 1d). As measured by genome-wide fixation index $F_{ST}$ (Fig. S3), Valdez and Haida Gwaii demonstrate levels of differentiation to SOCC and inland STOW that are similar to the differentiation between

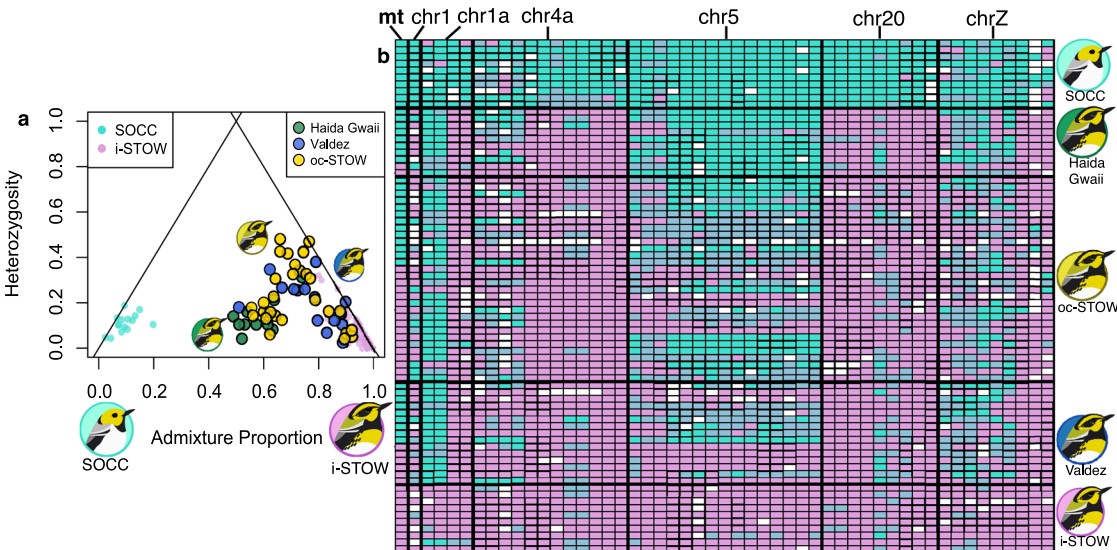

**Fig. 2 The signature of admixture in coastal S. townsendi (coastal STOW) populations. a** Triangle plot showing relationship between individual admixture proportion vs. heterozygosity based on 50 ancestry-informative SNPs ($F_{ST} > 0.6$) between *S. occidentalis* (SOCC) and inland *S. townsendi* (inland STOW), showing that coastal STOW populations (three genetic clusters in Fig. 1d, green: Haida Gwaii, blue: Valdez; army brown circling gold: oc-STOW) are admixtures between SOCC and inland STOW-like population. **b** Ancestry score (homozygotes SOCC and inland STOW, respectively, in turquoise and magenta, heterozygotes in light blue) of mtDNA and 50 ancestry-informative nDNA SNPs ($F_{ST} > 0.6$) in chr 1, 1a, 4a, 5, 20, and Z of randomly sampled 10 individuals from parental populations (SOCC and inland STOW), and different coastal STOW populations. Bird illustrations by Gil Jorge Barros Henriques.

SOCC and inland STOW. This is possibly a result of drift due to small local population size, especially on Haida Gwaii.

These data support the hybrid origin of coastal STOW proposed by ref. [29], as the coastal STOW population has mixed ancestry between SOCC and inland STOW, both at nDNA and mtDNA. At markers that are highly differentiated ($F_{ST} > 0.6$) between SOCC and inland STOW, many of the coastal STOW individuals have similar levels of heterozygosity as SOCC and inland STOW, indicating that hybridization first occurred many generations ago (Fig. 2, S2). This pattern is in contrast to what would be expected from recent, early generation hybrids, which would appear near the top of the triangle plot. The coastal STOW population falls towards the inland STOW end of the ancestry spectrum, indicating that the original admixture started with more inland STOW than SOCC or that there has been more introgression from inland STOW since the original admixture. Of those divergent SNPs included in the ancestry analysis, we found SOCC-biased introgression in restricted genomic regions on chromosomes (chr) 1A, 5, and Z (Fig. 2b).

**$F_{ST}$ distribution**. Genome-wide levels of differentiation show that SOCC and inland STOW demonstrate shallow genomic differentiation (Weir and Cockerham weighted $F_{ST} = 0.023$) except for a few peaks of differentiation (Fig. S1a). As in the PCA (Fig. 1d), $F_{ST}$ analysis indicates the Valdez, AK (USA) and Haida Gwaii, BC (Canada) coastal STOW populations are more differentiated from both SOCC and inland STOW than other coastal STOW are (see $F_{ST}$ values in Fig. S1, S3). The rest of the coastal STOW are more similar to inland STOW (Weir and Cockerham's $F_{ST} = 0.008$) than to SOCC (Weir and Cockerham's $F_{ST} = 0.015$) (Fig. S3).

The inland STOW and SOCC have a number of peaks of differentiation (Fig. S1a) mapping to chr 1, 1A, 4A, 5, 20, and Z in the *Setophaga coronata* reference. One of these (on chr20) is in the *ASIP-RALY* gene block[31], which is known to regulate both melanic and carotenoid pigmentation in quail and mice[35,36]. Our earlier study of admixture mapping in the ongoing hybrid zone between inland STOW and SOCC in the Washington Cascades[37] indicated that this locus is highly associated with plumage colour

patterns within that zone. As predicted, the present analysis of genomic variation over a much broader geographic region shows high differentiation at the *ASIP-RALY* SNP between sampling regions that differ in plumage (i.e., between SOCC and inland STOW, Fig. S1a, f–h) and low differentiation between regions with similar plumage (i.e., between coastal STOW and inland STOW, Fig. S1b–d).

Similar to the chr20 *ASIP-RALY* peak, the chr5 region also showed extreme differentiation in the comparison of inland STOW and SOCC (Fig. 3, S1), but in contrast to the *ASIP-RALY* region, the chr5 region is also strongly differentiated between coastal and inland STOW and is the most differentiated region between those groups (Fig. S1bc; 3bc). The chr5 peak (Fig. 3a–c) involves a ~1.2 Mb region—assayed here with 15 SNPs—and includes 22 genes. These genes are predominantly associated with energy-related signaling transduction (10/22) and fatty acid metabolism (7/22) (Supplementary Data 1). This chr5 peak was absent in Valdez (Fig. 3d), where there was less SOCC mtDNA introgression.

**Mitonuclear genetic association**. We then examined whether any nuclear genomic regions were correlated with mtDNA haplotypes (Fig. 1c), controlling for population substructure. This genome-wide association study (GWAS) revealed significant ($p_{corrected} < 10^{-6}$) mitonuclear covariation within chr5 (Fig. 4a). The mtDNA-associated SNP on chr5 was within the *PAPLN* gene (Fig. 4b, Supplementary Data 1), which resides within the island of differentiation between SOCC and inland STOW (Figs. 3a and 4b), as well as between coastal STOW and inland STOW (Figs. 3 and 4b). Since this ~1.2 Mb chr5 gene block is concentrated with genes with direct functional association with mitochondria (Supplementary Data 1), we thereafter considered this chr5 differentiation gene block (Fig. 4c) as a 'candidate gene block' for subsequent analysis. Among sites, the ancestry of the chr5 differentiation block (Fig. 4d, partial mantel Pearson's product-moment $r = 0.9227$, $p < 10^{-4}$) was correlated with mtDNA ancestry after controlling for spatial autocorrelation (see Methods). This association was not driven by highly differentiated

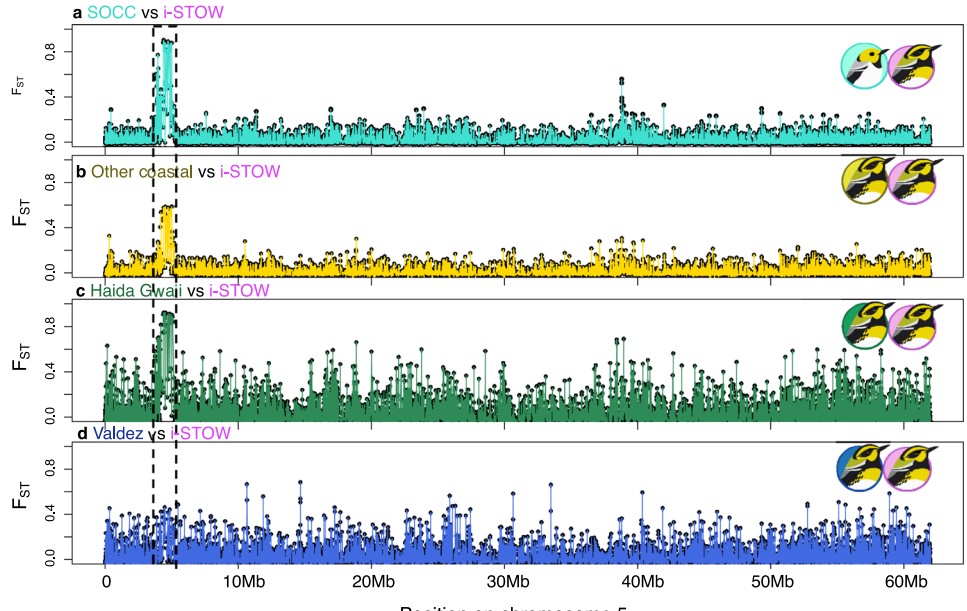

**Fig. 3 Semi-parallel differentiation of SOCC and various coastal STOW from inland STOW in a chr5 gene block (dotted box).** Genetic differentiation ($F_{ST}$) of variable sites along chromosome 5, between four pairs of populations, where *S. occidentalis* (SOCC), inland *S. townsendi* (i-STOW) were respectively color-represented by turquoise and magenta (**a**), and various coastal *S. townsendi* (coastal STOW) populations were color-represented in gold (**b** for other coastal STOW in Fig. 1d), forest green (**c** for Haida Gwaii coastal STOW), and royal blue (**d** for Valdez coastal STOW), respectively. Bird illustrations by Gil Jorge Barros Henriques.

coastal STOW (Valdez and Haida Gwaii), still showing strong site-level mitonuclear correlation (partial mantel Pearson's product-moment $r = 0.9568$, $p < 10^{-4}$) without those two sites.

**Signature of selective sweep.** Both the nucleotide diversity (π) and LD distributions revealed the expected signature of a selective sweep within the chr5 1.2 Mb gene block (Fig. 5). The mean of π within the candidate gene block was significantly lower than the rest of chr5 in coastal STOW (Fig. 5a, 95% CI of mean π, candidate block = 0.0011−0.0028, versus rest of chr5 = 0.0035 −0.0039). We focused on the region between 2.5 and 7 Mb within chr5 to examine a high-resolution LD landscape around the candidate genetic region (between ~3.73 and 5.17 Mb). LD was significantly higher within the candidate gene block than its surrounding genetic region (between 2.5 and 7 Mb) on chr5 in coastal STOW (Fig. 5b, 95% CI of mean $r^2$, candidate block 0.0883–0.0981, versus rest = 0.0232–0.0234). This expected signature of a selective sweep was evident in Haida Gwaii, Valdez, and the rest of coastal STOW (Fig. S4).

Genomic cline analysis revealed signatures of divergent selection on mtDNA and the associated chr5 gene block, as the β parameter estimates were significantly positive (Fig. 5c, d; Table S3). The divergent selection was asymmetrical, as the α is significantly negative (Fig. 5c, d; Table S3), suggesting excessive SOCC introgression, consistent with SOCC mitonuclear genes being selectively favored in the coastal STOW range.

**Climatic association.** Climate PC1 (Fig. 5e, S6a) explained 58.85% of the variation in climate among sites; this PC was negatively associated with temperature (Fig. S6a; Table S4). Climate PC2 explained 26.44% of the variation, mainly characterizing variation in precipitation among sites (Fig. S6a; Table S4). There was significant difference in climatic PC1 (Kruskal–Wallis chi-squared = 10.536, df = 2, $p = 0.0052$; pairwise Wilcoxon rank sum test with Benjamini–Hochberg $p_{corrected} < 0.05$) as well as PC2 (Kruskal–Wallis chi-squared = 16.405, df = 2, $p < 0.0003$)

among SOCC, coastal, and inland STOW breeding habitats (Fig. S6bc). Overall, the coastal STOW habitat is moister and cooler than both SOCC and inland STOW habitat (Fig. 5e, S6, Table S4), which is consistent with the latitudinal temperature gradient as well as the coast–inland humidity gradient in western North America. The distribution of mitonuclear ancestry (Fig. 5g) is significantly correlated with climate PC1 (Fig. 5e) (partial mantel test controlling for geographical distance, $r = 0.30$, $p = 0.0108$, Fig. 5e) as well as PC2 ($r = 0.20$, $p = 0.0464$, Fig. 5f) among 19 sites (across SOCC, coastal STOW, and inland STOW habitats).

**Discussion**

*Setophaga occidentalis* (SOCC) and inland *S. townsendi* (inland STOW) are distinct in mtDNA with signatures of positive selection on *ND2*. The nuclear genome exhibits 6 strong regions of differentiation in chr 1, 1A, 4A, 5, 20, and Z, whereas coastal *S. townsendi* (coastal STOW) harbor admixed mtDNA and nDNA ancestry from SOCC and inland STOW. One of the regions of strong differentiation between SOCC and inland STOW, a ~1.2 Mb region on chr5, differentiates coastal STOW and inland STOW as well. This gene block contains protein coding genes involved in NADH and ATP-associated reactions as well as mitochondrial fatty acid metabolism (Supplementary Data 1). Genetic variation in this region shows strong association with divergent mtDNA haplotypes, which indicates its co-adaptation with mtDNA. We found that site-level mitonuclear ancestry covaries with the site climate conditions (Fig. 5e–g), consistent with mitonuclear coadaptation to climate directly, or to other factors that are associated with climatic variation, such as forest characteristics.

We found that the key nDNA differences between coastal STOW versus inland STOW reside at a gene block within chr5, which covaries with mtDNA ancestry in coastal STOW. This region is functionally associated with fatty acid metabolism and energy-related signaling transduction (Fig. 4b; Supplementary Data 1), an intriguing result given that coastal STOW and inland

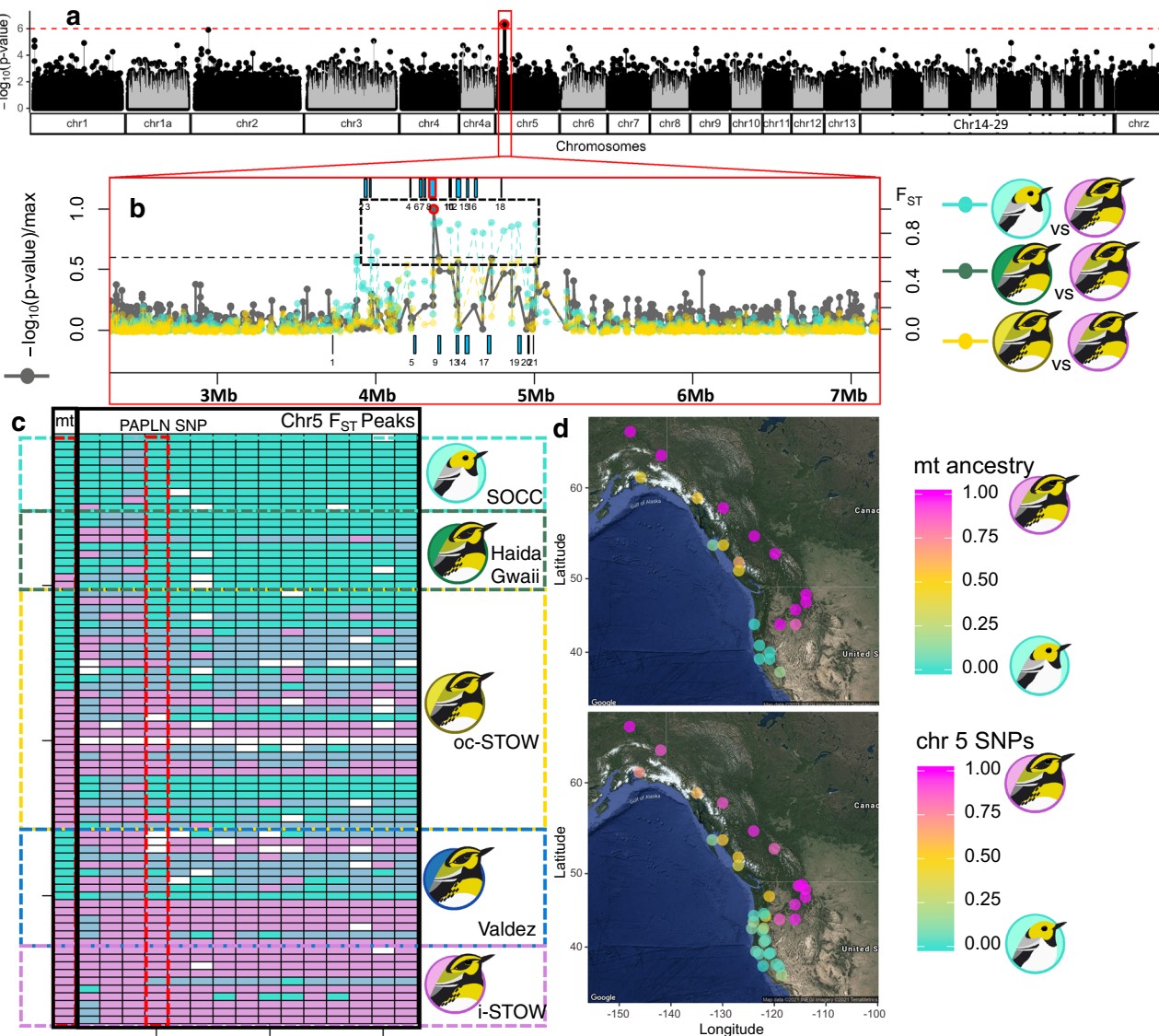

**Fig. 4 Mitonuclear association across SOCC, coastal STOW, and inland STOW populations. a–c** Mitonuclear association coincides with $F_{ST}$ peaks between *S. occidentalis* (SOCC) and inland *S. townsendi* (inland STOW), as well as between various coastal STOW versus inland STOW. **a** GWAS scan of mitonuclear association (controlling for population substructure) across the genome revealed significant associations ($p_{corrected} < 10^{-6}$ indicated by red circles) within chromosome 5. **b** The mitonuclear association (red circle) within chromosome 5 coincides with $F_{ST}$ peaks ($F_{ST} > 0.6$) between inland STOW (magenta logo) and SOCC (turquoise logo), between Haida Gwaii STOW (forest green logo) and inland STOW, as well as between other coastal STOW (yellow logo) and inland STOW. The colors of the $F_{ST}$ scans are consistent with the color logos of the populations that were compared with inland STOW. Blue boxes correspond to the positions of genes (top: forward strand; bottom: reverse strand) corresponding to the numbered IDs in Supplementary Data 1. The mitonuclear association peak is within Papilin, proteoglycan like sulfated glycoprotein (PAPLN) highlighted with red border. **c** Ancestry genotypes (turquoise = SOCC; light blue = heterozygous; magenta = inland STOW) of mitochondrial ND2 and chr5 SNPs correspond to strong genotypic differences between populations. Ten randomly sampled individuals from each parental population were shown along with the individuals from various coastal STOW. The PAPLN SNP genotypes are red bordered. **d** There is significant correlation between site mean ancestry of mtDNA (top, same as Fig. 1b) and chr5 (bottom) genetic block (shown in **c**) after controlling for geographical distance across sites (partial mantel test, $p < 10^{-4}$). Each data point represents a sampling site. Map data ©2019 Google. Bird illustrations by Gil Jorge Barros Henriques.

STOW differ so strongly in their mitochondrial haplotype frequencies. This chr5 gene block is concentrated (17/22) with genes encoding proteins directly involved in ATP and NADH activities, which corroborates with the four amino acid substitutions in the *ND2* gene and one in *ATP6* (Table S1, S2). In addition, the chr5 gene block harbors genes associated with fatty acid oxidation (Supplementary Data 1), which mainly occurs in mitochondria although it occurs in other cellular compartments as well (see reviews[38,39]). We further observed characteristics of a selective sweep within this gene block in coastal STOW along with

divergent selection on mitonuclear ancestry between SOCC and inland STOW (Fig. 5). Together, these findings point to mitonuclear coadaptation[11,12]. The SOCC nDNA may be partially incompatible with the inland STOW mtDNA in multiple functional roles, and vice versa, leading to selection against mismatched mitonuclear ancestries. Since the selection maintaining mitonuclear concordance is counteracted by random mating at each generation, it is likely only detectable when strong. Mitonuclear association both within and across sampling sites (Fig. 4) indicates strong selection maintaining a

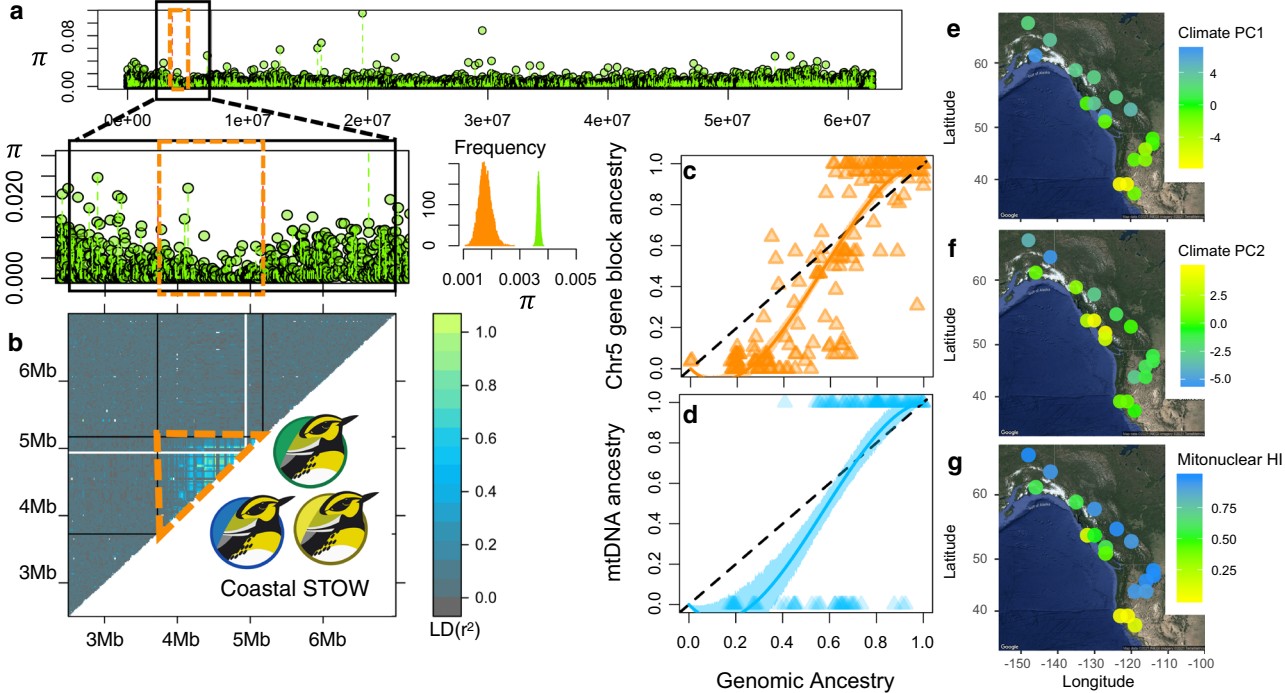

**Fig. 5 Signature of selection in mitonuclear ancestry is associated with climatic variation. a** There is lower nucleotide diversity (π) in the candidate gene block compared to the rest of chr5. The histogram illustrates the bootstrap distribution of mean π of the candidate gene block (orange) versus the rest of chr5 (green). **b** LD ($r^2$) was significantly higher in the candidate gene block (orange box) than its surrounding region between 2.5 and 7 Mb on chr5. Bird illustrations by Gil Jorge Barros Henriques. **c–d** Genomic clines of chr5 candidate gene block (**c**) and mtDNA (**d**), reflecting signatures of divergent selection as well as SOCC-biased introgression. Color shades represent the 95% confidence intervals of the genomic clines. **e–f** Spatial distributions of climate PC1 and PC2. **g** Site mean mitonuclear ancestry (averaged ancestry of mtDNA and chr5 candidate gene block) is correlated with local climate PC1 (partial mantel test, $r = 0.30$, $p = 0.0108$), as well as PC2 (partial mantel test, $r = 0.20$, $p = 0.0464$). **e–g** Map data ©2019 Google.

functionally compatible mitonuclear metabolic function over a large spatial scale.

Mitonuclear ancestry variation is significantly associated with climate variation among sites (Fig. 5e, f) that are divergent in climatic conditions (Fig. S6), suggesting that the divergent selection on the mitonuclear ancestry might be driven by climate and/or habitat (which is also associated with climate). The climatic condition in the coastal STOW habitat is moister than that in the inland STOW habitat (Fig. S6). These patterns are reminiscent of the *Eopsaltria australis* (Eastern Yellow Robin) system in which distinct mt-nDNA combinations are maintained between inland and coastal climatic conditions[6]. Fatty acid metabolic genes have also been shown to be targets of climatic adaptation in humans, within Siberian[40] and Greenlandic Inuit populations[41]. Temperature[42] and humidity[43] both influence mitochondrial fatty acid metabolism[42,43]. Chr5-mtDNA genotypes might result in functional difference in fatty acid metabolism that is adapted to specific climatic condition (cool and moist versus hot and dry) in the breeding habitat of these warblers.

Because SOCC has apparently inhabited coastal areas for a long period of time[28], the SOCC mitonuclear gene combination may be more suited for coastal habitats compared to those of inland STOW. The SOCC-biased mitonuclear genomic cline (Fig. 5c, d) supports this possibility. If the SOCC mitonuclear genotype is favored in the coastal habitats, the frequency of SOCC mt-nDNA gene combinations would tend to increase in coastal STOW populations over time. However, ongoing gene flow between inland STOW and coastal STOW would slow down or prevent such increase. The Haida Gwaii and Valdez populations could have escaped from such a balance between selection and gene flow due to their isolation from the rest of the populations, respectively, by the sea and mountain ranges. Although we could

not retrace the SOCC mitonuclear ancestry change along with climate change post-glacial-maxima, continuous tracking of coastal STOW population might reveal increase of SOCC ancestry as contemporary climate warming continues.

Another possibility is that frequency-dependent selection is maintaining long-term mt-nDNA polymorphism in the coastal STOW habitat, which is climatically distinct from both SOCC and inland STOW (Fig. S6). While the SOCC habitat is moister and warmer than inland STOW habitat, the coastal STOW habitat is colder than SOCC habitat, while moister than inland STOW habitat. Such a unique habitat climatic combination could divergently select for mitonuclear metabolic combinations that are distinct from both of the parental populations. Future investigation on the spatial and temporal variation of the strength of association between mtDNA and the chr5 gene block would shed light on the evolutionary forces shaping the present and future of the coastal STOW population. Further examining such mitonuclear relationships in the three hybrid zones between SOCC and inland STOW in Washington and Oregon[27] would shed light on the prevalence of such evolutionary forces.

In comparisons among these populations, the distribution of $F_{ST}$ across the genome is consistent with the "genic" view of differentiation[31,44,45], in which peaks of differentiation represent genetic targets of selection (divergent selection or selection against hybrids) that are highly distinct between populations despite the rest of the genome being homogenized by gene flow[44–46]. Despite this 'selection with gene flow' scenario exhibiting a similar genomic differentiation landscape as the classic 'divergence with gene flow' model[45,47], the underlying process is different. In this system, there is a known allopatric phase when SOCC and inland STOW were separated by ice sheets (Fig. 1a)[13,29]. Genetic differentiation that accumulated in

allopatry (as opposed to gradual build up at sympatry or parapatry under 'divergence with gene flow') can be homogenized by hybridization at secondary contact, while the climate/habitat-related genomic targets of selection (e.g., on chr5) remain differentiated.

In addition to providing an empirical examination of mitonuclear coevolution, the present study helps clarify the biogeographic history of this warbler complex. Our genomic evidence is consistent with Krosby and Rohwer's[29] conclusion, based on mtDNA, that coastal British Columbia and Alaska was inhabited by geographically structured SOCC populations before inland STOW expanded from inland areas and mixed with them[29]. The SOCC and inland STOW mtDNA haplotype groups demonstrate many fixed differences including 5 nonsynonymous substitutions in ND2 and ATP6 (Table S1), of which 2 result in physiochemical changes, whereas both haplotypes are common in coastal STOW. It is unlikely that the polymorphisms in mtDNA and nDNA in the coastal STOW were caused by incomplete lineage sorting, as opposed to hybridization (Fig. 1a). In a scenario of incomplete lineage sorting, it is unlikely that both inland STOW and SOCC would have completely lost alternative haplotypes while the coastal STOW maintained both. If a 50% ancestral inland STOW population began with the SOCC haplotype, under the Wright–Fisher model[48–50], based on an effective population size of $3.4 \times 10^5$ ($\frac{\pi}{4\mu}$, where $\pi$ is 0.0015[51], $\mu$ is $4.42 \times 10^{-9}$ per generation[52]), it would take around $9.4 \times 10^5$ generations to completely lose the SOCC haplotype. Over such prolonged period, sizeable differences would be expected between the mtDNA haplotypes found in SOCC, coastal STOW, as well as inland STOW populations. We did not observe such a pattern, as some coastal STOW mtDNA haplotypes were identical to SOCC haplotypes and some were identical to inland STOW haplotypes.

The higher genome-wide differentiation of the Haida Gwaii and Valdez populations (Fig. S1, S3) is consistent with at least partially isolated cryptic refugia of SOCC in coastal Alaska and Haida Gwaii during the last glacial maximum[53]. Following expansion of inland STOW from the inland area with the boreal forest post-glacial expansion around 5000 years ago[54,55], hybridization between inland STOW and SOCC apparently led to populations of mixed ancestry along the coast of British Columbia and Alaska (Fig. 1a). These coastal populations have the plumage patterns and colors of inland STOW, which is why they have been classified as members of that species. This uniform inland STOW appearance has concealed a more complex history of hybridization with geographically differentiated populations of SOCC.

Following expansion of inland STOW from the interior, gene flow into Haida Gwaii may have been weak due to the expanse of water separating it from the mainland, explaining why that population is more similar to SOCC than other coastal STOW are. Gene flow into Valdez could have also been impeded by geographical barriers, as Valdez is surrounded by mountain ranges (Chugach mountains, Wrangell mountains, and St. Elias mountains). However, both mitochondrial and nuclear genomic data indicate that Valdez has substantial ancestry from both inland STOW and SOCC. Despite genome-wide differentiation among the three coastal STOW genetic clusters (Fig. 1d), there is an interesting parallelism: all the three populations exhibit disproportionately higher frequencies of the SOCC-like chr5 nDNA gene block that covaries with mtDNA (Fig. 3–4). Such parallelism sheds light on potential parallel adaptation to the coastal climate, which warrants further testing.

While our findings are consistent with mitonuclear coadaptation being important in the pattern of genomic differentiation within this species complex, this being an observational study we

cannot definitively conclude that is the case. The mitonuclear association, signatures of selective sweep within the mtDNA-associated nuclear candidate gene block with genes directly involved in mitochondrial functions, and the correlation of mitonuclear ancestry to climatic conditions together build strong correlative evidence for mitonuclear co-adaptation. One possibility is that the nuclear and mitochondrial loci are independently selected by the environment, without actual coevolution between the two. Future study should investigate this possibility to distinguish it from actual coevolution.

Examination of genomic differentiation in this young species complex has revealed patterns consistent with climate-related coadaptation among mtDNA and nDNA involved in energy-related signaling transduction and fatty-acid metabolism. Coastal STOW demonstrate both mtDNA and nuclear genomic patterns consistent with admixture between past SOCC and inland STOW populations. Three genetic clusters of coastal STOW are characterized by a mixed genetic ancestry between the parental populations (SOCC and inland STOW), providing natural replicates for examining the role of selection in shaping genomic differentiation. These three coastal STOW clusters exhibit differentiation from inland STOW at one of the most differentiated genomic regions (on chr5) between inland STOW and SOCC, which demonstrates signatures of divergent selection. The geographic distributions of the mitonuclear genetic combinations related to NADH and ATP-related reactions are associated with geographic variation in climate, suggesting mitonuclear coevolution may have occurred in response to selection for climate adaptation. Such climate-related mitonuclear selection could be an important force driving population differentiation.

## Methods

**Museum samples, mtDNA sequences, and nDNA sequencing**. As a baseline for understanding the relationships among mtDNA haplotypes and their distributions, sequences of the mtDNA *NADH dehydrogenase subunit 2* (*ND2*) gene for 223 individuals (95 coastal STOW, 81 inland STOW, and 47 SOCC) from the Krosby and Rohwer[27] study were acquired from GenBank (accession numbers FJ373895-FJ374120). To further understand the relationships among various STOW populations with these mtDNA sequences, we generated a minimum spanning haplotype network[56] with PopART[57]. This network showed two clearly separated haplotype clusters. We then scored each haplotype as 0 for those within the SOCC haplotype cluster and 1 for those within the inland STOW cluster[29] (Fig. 1c).

From samples with previously sequenced mtDNA (i.e., from ref. [29]), we selected a subset of muscle tissue samples (70 inland STOW, 57 coastal STOW, and 15 SOCC; obtained from the Burke Museum of Natural History and Culture, University of Washington, Seattle, Washington) for nuclear genomic sequencing. We supplemented this set of genetic samples with 54 blood samples that we obtained directly from birds caught in the field during the breeding season of 2016; these included 47 SOCC from California, 5 inland STOW from Montana, and 2 inland STOW from Washington. Therefore, 196 samples (62 SOCC, 77 inland STOW, and 57 coastal STOW) were included in the nuclear genomic sequencing. Both the museum muscle tissue samples and new blood samples yielded sufficient DNA, and there was not a noticeable difference in sequence data between the muscle and blood samples sampled from similar geographical areas. Animal sampling and handling complies with ethical guidelines provided by Environment Canada, U. S. Geological Survey, Departments of Fish & Wildlife of Washington, California, and Montana, and the Animal Care Committee of University of British Columbia.

**GBS pipeline**. We prepared genotyping-by-sequencing (GBS)[58] libraries from DNA samples of the 196 individuals described above as our previous study[37]. Briefly, we digested the genomes with the restriction enzyme PstI, then ligated fragments with barcode and adaptors, and amplified with PCR. Adaptor primers are listed in Table S5. Amplified DNA was pooled into two libraries which were then paired-end sequenced at Genome Quebec: the first (115 individuals) were sequenced with an Illumina HiSeq 2500 automated sequencer (read length = 125 bp), and the second (81 individuals) were sequenced with an Illumina HiSeq 4000 (read length = 100 bp) due to equipment updates at the sequencing facility. We randomly assigned samples to different plates and included replicates of three samples among plates. Sequence processing was based on that of a previous study[37], but with some changes included in the summary below. Specifically, the reads were demultiplexed with a custom script and then trimmed using Trimmomatic[59] [TRAILING:3 SLIDINGWINDOW:4:10 MINLEN:30]. We aligned

reads to a congener *Setophaga coronata*[60] with bwa-mem[61] (default settings). Variable sites were identified with GATK[62], then filtered with VCFtools[63] according to the following criteria: (1) removing indels; (2) keeping sites with genotype quality (GQ) > 20; (3) keeping sites with minor allele frequency (MAF) ≥ 0.05; (4) removing sites with >30% missing genotypes among individuals; and (5) keeping biallelic SNPs only. Thereafter 222,559 SNPs remained.

**Population structure and genomic differentiation**. To investigate if the coastal STOW is a product of genetic admixture of SOCC and inland STOW, we first examined population structure using PCA in the SNPRelate[64] package in R[65], followed by heterozygosity-ancestry proportion analysis[66]. We originally set out to assess the differences between SOCC, coastal STOW, and inland STOW. However, the PCA revealed strong clustering within coastal STOW with Valdez and Haida Gwaii populations distinct from the rest of the coastal STOW populations. In subsequent analysis, we thus compared each of the three coastal STOW groups (Haida Gwaii, Valdez, and other coastal STOW [oc-STOW]) to the inland STOW and SOCC groups. To examine differentiation among hybrid and parental populations (Fig. S3) across the genome, for each of the 222,559 filtered SNPs we calculated $F_{ST}$[67] with VCFtools[63] between (1) inland STOW ($n = 77$) and SOCC ($n = 62$); (2) coastal STOW ($n = 57$: 10 Haida Gwaii, 15 Valdez, 32 others) and inland STOW; and (3) SOCC and each of the three coastal STOW clusters.

Extended generations of hybridization are expected to result in individuals with variable ancestry and reduced heterozygosity, in contrast to recent, early generation hybrids, which will be heterozygous across most differentiated loci. To uncover signatures of extended hybridization, we calculated ancestry and inter-specific heterozygosity of the coastal STOW individuals. We considered SNPs with $F_{ST} > 0.6$ (between inland STOW and SOCC) as being useful for informative parental ancestry. There were 50 such SNPs distributed across chr 1, 1A, 4A, 5, 20, and Z. For each SNP, we assigned ancestry scores of 0 and 1 for homozygous SOCC and homozygous inland STOW variants, respectively, and 0.5 for heterozygotes. We calculated ancestry (averaged ancestry score of the 50 SNPs) and heterozygosity of the 50 SNPs with $F_{ST} > 0.6$. To ensure that neutral sites nearby selected genomic regions were considered, we repeated heterozygosity-ancestry analysis with a $F_{ST}$ threshold of 0.3, which involves 618 SNPs.

Out of the 50 high-$F_{ST}$ SNPs (from above), 15 were on chr5 (Figs. 3 and 4b, c). These are all within an $F_{ST}$ peak between coastal STOW and inland STOW, and also within $F_{ST}$ peaks between SOCC and inland STOW.

**Association of mtDNA and nDNA**. To examine within-population association between mtDNA and nDNA ancestry, we conducted a GWAS with GenABEL[68] in R to examine if there is association between mtDNA group (0 for SOCC type, or 1 for STOW type) and nuclear genotypes across the genome while controlling for population substructure. We used the egscore function that tests association between mtDNA ancestry and nuclear SNP ancestry while controlling for population substructure using the identity-by-state kinship matrix (calculated by the function ibs)[68]. We subsequently conducted genomic control assuming that randomly selected sites are not associated with mtDNA. To do so, we calculated inflation factor $\lambda$ (accounting for both the effects of genetic structure and sample size) and used it to correct for the test statistic $\chi^2$ (and thus $p$-values) of each mitonuclear association test. A $p$-value cutoff of $10^{-6}$ was employed to account for multiple hypothesis testing for sites across the genome as a balance of Bonferroni correction (0.05/222559 SNPs = $2.2 \times 10^{-7}$) and dependency among linked SNPs. To examine gene function related to the loci associated with mtDNA haplotypes, we examined known protein-coding genes within the chr5 $F_{ST}$ peak delineated by the 15 SNPs (see above) including the candidate SNP with $p$-value <$10^{-6}$, using the *S. coronata* annotation[60]. For each gene, we identified the *Taeniopygia guttata* homolog[69], which was then searched for Gene Ontology molecular and biological functions with UniProt[70].

We took the union of the $p < 10^{-6}$ SNPs (see GWAS above) and the $F_{ST}$ peak between coastal and inland population, which resulted in 15 SNPs in a 1.2 Mb block on chr5 that we refer to as the 'candidate gene block'. We further tested whether this candidate gene block covaries with mtDNA ancestry across coastal STOW sites. We first calculated nDNA ancestry for each individual by averaging the locus-specific ancestry within the 1.2 Mb candidate gene block within chr5 (0 for homozygous SOCC, 0.5 for heterozygotes, 1 for homozygous inland STOW). Then we calculated mitonuclear ancestry by averaging mtDNA ancestry (see above) and nDNA ancestry for each individual. Finally, we calculated the mean mitonuclear ancestry of each site by averaging mtDNA and nDNA ancestries. To examine between-population association between mtDNA and nDNA variants of interest, we employed a partial mantel test[71] with the vegan package in R to quantify the association between the among-sites ($N = 19$, the intersect of sites of which we have mtDNA and nDNA ancestry estimates) distance matrices of mtDNA ancestry and the nDNA ancestry, while controlling for geographical distance. Among the 19 sites, the number of SOCC, coastal STOW, and inland STOW sites were, respectively, 6, 6, and 10. The high differentiation of Valdez and Haida Gwaii from other coastal STOW could disrupt the continuous spatial distribution expected by the partial mantel test. Therefore, we tested site-level mitonuclear association with/without these two sites to account for any potential statistical artifacts.

**Signature of selective sweep within chr5 candidate region**. To test for signatures of a selective sweep within the candidate gene block in the coastal STOW populations, we calculated nucleotide diversity ($\pi$) with pixy[72] and linkage equilibrium (LD) around the ~1.2 Mb candidate gene block (see above) with VCFtools and the input being all the coast STOW samples. We calculated pair-wise LD as the squared correlation coefficient of genotypes with the '—geno-r2' command. For nucleotide diversity analysis, we excluded the minor allele frequency filter (step (3) above), which resulted in 1,022,654 total sites (SNPs along with invariant sites). We calculated unbiased $\pi$ with non-overlapping windows of size = 1 kb with pixy[72] along positions in the reference genome. To test if there was significant reduction in $\pi$ and elevation in LD within the candidate gene block compared to its flanking region between 2.5 Mb and 7 Mb on chr5, we bootstrapped $\pi$ and LD within the candidate gene block and its flanking region on chr5 for 10,000 iterations. We compared the 95% CIs of mean $\pi$ for the candidate gene block versus mean $\pi$ in the rest of chr5, as well as 95% CIs of mean for the candidate gene block versus its flanking region between 2.5 Mb and 7 Mb on chr5. To examine repeatability among coastal STOW populations, we also performed the $\pi$ and LD analysis within each coastal STOW genetic cluster (Haida Gwaii, Valdez, and the rest coastal STOW) seperately.

To further understand the direction and strength of selection on mtDNA and the candidate gene block, we used a genomic cline model[73]. Specifically, we fit the genomic cline function: $\theta = h + (2\,(h - h^2) \times (\alpha + (\beta\,(2\,h))$, where $\theta$ is ancestry of the chr5 candidate gene block (see above) and $h$ is the genome-wide ancestry for each individual. The genome-wide ancestry is estimated as a transformed genomic eigenvector 1 (EV1) such that $h$ varies from 0 (SOCC) and 1 (inland STOW), respectively: $h = (EV1 - \text{minimum}(EV1)) / (\text{maximum}(EV1) - \text{minimum}(EV1))$. Fitting this model allows estimation of $\alpha$ and $\beta$ parameters that represent deviations of the cline of a specific locus from the genomic cline center and shape, respectively. If the chr5 candidate gene block is neutral in its association with the fitness of admixed individuals, we expect $\alpha$ and $\beta$ to be 0. If the chr5 gene block is under divergent selection, $\beta$ should be positive. SOCC-biased or inland STOW-biased introgression within the chr5 gene block would, respectively, correspond to a negative or positive $\alpha$. To estimate $\alpha$ and $\beta$ parameters, we employed nonlinear weighted least-square regression with the nls function in stats package[74] in R, where we adopted the genomic cline function into the nls function: $\theta = h + (2(h - h^2) \times (\alpha + (\beta(2\,h) - 1)) + \epsilon$, where $\epsilon$ is the residual.

**Analysis of mtDNA sequence**. To infer potential functional underpinnings of nDNA–mtDNA interactions, we compared sequence divergence of protein coding genes in full mitochondrial genomes of SOCC and STOW. We used published whole-genome data from tissues sequenced by Baiz et al.[57] to extract high quality (>1000X coverage) mtDNA reads ($n = 4$ SOCC haplotypes, $n = 6$ STOW haplotypes; see ref. [60] for details on library preparation, sequencing methodology, and sample information). Reads were aligned to the *S. coronata* mitochondrial genome, which we annotated using the MITOS[75] web server annotation platform. To generate individual mtDNA fasta files, we used the "mpileup" command in samtools version 0.1.18[61], the "call -c" command in bcftools version 1.10.2, and the "vcf2fq" option to generate fastq files. Finally, we used the program seqtk (version 1.3; https://github.com/lh3/seqtk) to translate into fasta format. We generated sequence alignments, calculated the number of fixed SNPs between the divergent haplotypes, and determined the number of fixed amino acid substitutions in Geneious[76] (11.0.3).

To assess the physicochemical impacts of the substitutions, we analyzed the sequences using TreeSAAP v3.2[77]. We assessed the 23 TreeSAAP physicochemical properties with reported accuracies >70%[78] and considered radical changes (category 6–8) at $p < 0.001$ to show evidence for positive selection. To further explore the impact of each substitution, we assessed the mutations using Provean[32], which predicts the functional relevance of a substitution based on sequence conservation amongst homologous sequences. Additionally, we used Mixed Effects Model of Evolution (MEME)[34] within the Hyphy suite to test for episodic diversifying selection of the *ND2* and *ATP6* sequences, supplementing the STOW and SOCC haplotypes with an additional 80 *ATP6* and 101 *ND2* sequences from Parulidae and Icteridae (Supplementary Data 2).

**Climate analysis**. To investigate whether there might be selection on mt-nDNA related to climate, we tested association of site-level mt-nDNA ancestry (the averaged site ancestry score of mtDNA and chr5 candidate gene block ancestry) and climate variation. To effectively capture annual climate variation among sites, we extracted data of 26 climate variables (Table S4) from ClimateWNA[79], which is based on climate data collected from 1961 to 1990, with a resolution of 2 decimal places of longitude and latitude. We used PCA to reduce the dimensions of climatic variation among sites. To test if there was difference in PC1 or PC2 among SOCC, coastal STOW, and inland STOW habitats, we employed a Kruskal–Wallis test followed by a pairwise Wilcoxon rank sum test with Benjamini–Hochberg multiple hypothesis correction. We computed pairwise differences between sites for (a) climate based on PC1 or PC2, (b) geographic distance, and (c) mitonuclear ancestry. The site mitonuclear ancestry was the average of the mtDNA ancestry and ancestry of the chr5 candidate gene block (see above) of each sampling site. We then tested for an association between climate differences among sites and

differences in mt-nDNA ancestry while controlling for geographic distance using a partial mantel test[80] in R with 10,000 permutations.

**Reporting summary**. Further information on research design is available in the Nature Research Reporting Summary linked to this article.

## Data availability

The sequence data generated in this study have been deposited in the GenBank SRA database under accession code PRJNA573930 and PRJNA642412. The processed genomic data are available at Dryad (https://doi.org/10.5061/dryad.44j0zpc9t).

## Code availability

Code (DOI:10.5281/zenodo.4782431) involved in this paper is deposited in Github (https://github.com/setophaga/codes-for-Evo-manuscripts/blob/master/Mitonuclear.warbler.div/).

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

## Acknowledgements

In support of the Bird Names for Birds movement, we omitted the existing common name of *Setophaga townsendi*. We are grateful to Sharon Birks (Burke Museum) and Chris Wood (Burke Museum) for access to the tissue samples for sequencing. We thank Geoffrey E. Hill for inspiring ideas to this study. We also thank Gil Henriques for providing digital illustrations of the warblers. For helpful discussion we thank Sally Otto, Dolph Schluter, Loren Rieseberg, Graham Coop, Dahong Chen, Mike Whitlock, Andrea Thomaz, Armando Geraldes, Meade Krosby, Hernan Morales, Xinzhu Wei, Jared Grummer, Rohit Kolora, and Jessica Irwin. We are grateful for research funding provided by the Natural Sciences and Engineering Research Council of Canada (grants 311931-2012, RGPIN-2017-03919 and RGPAS-2017-507830 to DEI; and PGS D 331015731 to SW); Pennsylvania State University, the Eberly College of Science, and the Huck Institutes of the Life Sciences to DPLT; and a Werner and Hildegard Hesse Research Award in Ornithology and a UBC Four Year Doctoral Fellowship to SW. For research permits we thank Environment Canada; U. S. Geological Survey; Departments of Fish & Wildlife of Washington, Idaho, California, and Montana; and the UBC Animal Care Committee.

## Author contributions

S.W. and D.I. designed the study. Museum samples were provided by S.R., while D.I. provided funding and equipment. S.W. and E.K.M. collected field samples. S.W. prepared museum samples. E.K.M. and S.W. extracted DNA. With guidance from S.W., M.J.O. prepared libraries for sequencing. S.W. conducted all the data analysis, except the mitochondrial functional analyses that E.K.M. and D.P.L.T contributed. J.L.-Y. provided code for extracting climate data for sampling sites of interest. S.W. wrote the first draft and worked with D.I. before all authors contributed towards the final version of this paper.

## Competing interests

The authors declare no competing interests.
