## [Peer Review File · Nature Communications]

REVIEWER COMMENTS

Reviewer #1 (Remarks to the Author):

I found this to be a very important study. Along with a few other recently published papers on introgression of mt genes and mitonuclear coadaptation, this study holds the potential to reshape how evolutionary biologists think about the process of speciation.

The authors present results from genetic and statistical analyses on the nuclear and mitochondrial ancestry of two warbler species and subpopulations within those species. They find compelling evidence for ancient hybridization between ancestral populations of each species, resulting in intermediate populations with mixed ancestry. Scans of the nuclear genome revealed islands of differentiation among species and subpopulations that are associated with genes known to encode products important for mitochondrial energy metabolism. Thus, the authors suggest mitonuclear effects might have played important roles in shaping current genetic structure and plumage differences among these North American Warblers. The study seems thorough and the manuscript is well-written. I enjoyed reading the paper and I think the study will have broad implications toward understanding speciation.

I do have some suggestions for improving the paper. The bulk of these rest in interpretation and use of PCA components for correlational analyses on selection.

Major concerns:

Lines 213-219: The use of PCAs for further correlations is tricky and it is important to make expressly clear what the PC axes used for correlations represent. PCAs are most effective when used to cluster groups based on many variables in order to produce a "caricature" that is an amalgam of the variables included in the PCA. For example, a PCA of height, weight, and BMI can produce a PC1 that might represent how "fat" or how "muscular" an individual is, depending on the significance and sign (+/-) of the PC loadings.

What did PC1 and PC2 represent in the PCA performed here? (ie., Does PC1 represent how hot/dry a climate is? Does PC2 represent how cold/humid a location is?) If the PC loadings are too complicated to interpret this way, then it may be a sign that biological interpretation is lost, or that too many variables are included in the PCA to allow for biological inference from further correlations. This is why using PCAs in this way has fallen out of favor with some biologists (see Connor 2007, Chong 2018, and Hunt 2007 for more details, explanations and suggestions). I've studied table S4, and I'm not sure it is possible to get an accurate picture of what the climate PCA actually represents about climate based on those loadings. I suggest that the authors explain to the reader more clearly what is represented by PC1 and PC2.

I realize this is a big ask at this stage, but I suggest that the authors consider reducing the number of climate variables to a smaller group likely to affect energetic adaptation (such as temperature based on results like Healy 2019), or to key variables meant to specifically encompass key differences between coastal and inland habitat (forestation, temperature, humidity). Narrowing the focus beforehand based on biological inference may inform interpretation of results after the fact. Alternatively, the authors might better justify the way they employed PCA.

Lines 297-299: The breakdown of PC2 comes close to giving us a picture of what this PC represents. What were the signs of TD, CMI, PPT_wt, and MAP? If there were all positive, does this make PC2 represent how humid, warm, rainy and wet during the winter a location is? If this is correct then a better explanation here goes a long way toward addressing my concern in Lines 213-219 above.

Line 302-303: "Overall, coastal STOW habitat is moister and more stable in temperature" According to the PC loadings, or qualitative assessment of the climate outside of stats? Very clear descriptions would be helpful here.

Minor comments:

Line 46: would "isolated" or "allopatric" be better here?

Line 53: North American*

Lines 73-76: I think I understand what is being suggested here, but the language could be clearer. The assumption is that inland STOW and SOCC mtDNA haplotypes exist in an unbalanced frequency in coastal STOW based on location. This variation in frequency from location to location might spur differentiation WITHIN the coastal STOW subpopulations - presumably based on local adaptation? Is this the current hypothesis?

Line 82-82: For goal 3, is this within just coastal STOW? or across all groups in this study? I'm not sure this becomes clearer after reading the discussion either

Line 96: What tissue was used for these samples?

Line 102: Any noticeable differences among old frozen samples and new blood samples?

line 126: examined*

Line 195: nDNA*

In methods: It is a matter of personal preference, but I argue that software (including R packages) should be cited in manuscripts. Software racks up thousands of citations, but some people utilize functions in academic search engines to see what kind of studies software is currently being used for. This not only helps other researchers decide whether or not to use a package, but it also can increase traffic to studies that cite software.

Line 263: melanin pigmentation? Carotenoid pigmentation? STOW and SOCC differ in both.

Literature cited in this review

Conner, J. K. (2007). A tale of two methods: putting biology before statistics in the study of phenotypic evolution. *Journal of Evolutionary Biology*, 20(1), 17-19.

Chong, V. K., Fung, H. F., & Stinchcombe, J. R. (2018). A note on measuring natural selection on principal

component scores. *Evolution letters*, 2(4), 272-280.

Harada, A. E., Healy, T. M., & Burton, R. S. (2019). Variation in thermal tolerance and its relationship to mitochondrial function across populations of *Tigriopus californicus*. *Frontiers in physiology*, 10, 213.

Hunt, J., Wolf, J. B., & Moore, A. J. (2007). The biology of multivariate evolution. *Journal of evolutionary biology*, 20(1), 24-27.

Reviewer #2 (Remarks to the Author):

The paper "Signatures of mitonuclear coevolution in a warbler species complex" explores a hot topic in evolutionary biology: coevolved mitonuclear genotypes as a mechanism of climate adaptation. This is done in a system, where hybridization between two bird species that are phenotypically distinct but genetically similar resulted in divergent mitochondrial haplotypes co-existing across the broad coastal hybrid zone. Authors tested whether some nuclear genes show differentiation according to the mitochondrial haplotypes, as would be expected under selection for co-evolved mito-nuclear allele combinations to maintain proper mitochondrial function. They found a small genomic region on Chromosome 5, enriched with genes associated with mitochondrial function, with alleles strongly correlated with mtDNA haplotype. Variation at one nDNA SNP in that region had a near-perfect correlation with mtDNA haplotype; this SNP was located in PAPLN gene with function in energy-related signalling transduction, including in mitochondria. Together with some evidence for correlation between climatic and mtDNA variation, these results are used to conclude that adaptation to environment drives mitonuclear differentiation and introgression.

The evidence for selection for co-function of mtDNA and nuclear gene block on Chr5 appears strong, with only few mismatches between mtDNA and PAPLN SNP observed in the data (Fig 4). But the evidence for environmental selection for different mitochondrial function (i.e. better fit of the SOCC-like mitonuclear genotype to coastal climate; L361) is not so convincing. If hybridization is ancient (e.g. L13, 414), and climatic selection on a mitonuclear genotype is current, then why selection did not remove the less fit mitonuclear genotype? Significant correlation between climate PC1 and mitonuclear genotype appears to be driven by similarity between iSTOW and cSTOW, whereas climate PC2, for which cSTOW and SOCC were similar, was not significantly associated with mitonuclear ancestry. This does not support the better fit of the SOCC-like mitonuclear genotype to coastal environments. Is it possible that selection against less fit iSTOW-like mitonuclear genotype was historically strong but is currently weak due to post-LGM changes in climate? Are estimates of gene flow between other iSTOW and cSTOW available to support or refute the idea that ongoing gene flow supplements coastal populations with iSTOW alleles (L364)? Because Valdez and Haida Gwaii, which are said to be more sheltered from current gene flow from iSTOW, also have both mtDNA haplotypes, selection for SOCC-like haplotypes in coastal environments does not appear likely. Thus, I suggest to tone down inferences of climate-related selection and adaptation.

It was surprising that such low mitochondrial divergence (0.8%, 5 fixed amino acid substitutions) can

potentially result in strong mitonuclear incompatibilities- is there any support for the mitochondrial haplotypes (and potentially respective mitonuclear genes) being functionally different, e.g. based on physicochemical properties (TREESAAP, Woolley et al. 2003) or functional analyses (PROVEAN, Choi et al. 2012)? Valdez population did not have a Chr5 peak when compared to iSTOW (Fig. 3, S1)- why could this be? Genomic FST peaks could arise due to low level of variation in the region- does FST peak on Chr5 also correspond to a peak in average sequence divergence in this region, as would be expected under divergent selection for different mitonuclear combinations? Or, could the FST peak on Chr5 (and its correlation with mtDNA) result from sex-linkage?

My other comments are minor, mainly on clarity of expression and better transparency of methods and results. The paper is well written, methodologically sound and is a pleasure to read. This work will make a significant contribution to mito-nuclear coevolution and speciation research.

Other comments:

L10-12 please split this sentence. You probably don't have to abbreviate inland and coastal STOW in the abstract.

L14 Perhaps start this sentence with "One of the few highly-differentiated nDNA regions...". Also, please rephrase to clarify what covaries with what (presumably, variation in nDNA with variation in mtDNA).

Same comment for L20, L316: a region cannot correlate with mtDNA haplotypes, variation within a region can.

L29-32 Split this sentence. In L31 specify that you mean the mitochondrial and nuclear genomes.

L49 do you mean selection AGAINST suboptimal combinations of ALLELES?

L54 add "coastal" after "inhabits"?

L58 move (abbreviated as inland STOW)" to L56 after "inland population".

L70 add "genomic" before "regions" (unless you mean geographic?)

L71 ref 30 is a bioRxiv version of 29- is this an error? Same in L256

L77-83 The section describing author's motivation for the study needs strengthening. Why would incompatibilities be expected in a stable hybrid zone, where mtDNA haplotypes are only 0.8% different? Can Question 2 be answered with current correlative approaches? It might be better to formulate hypotheses about drivers of the observed patterns, and seek support for these hypotheses. It could be also helpful to briefly summarize the approaches taken to address these questions in the end of the Introduction.

L134 It was unclear what "population" refers to here- are populations based on nuclear genetics or geography? It was also not clear what you call sites and how many of them are plotted on the maps- it would be good to have a table somewhere in Supp Mat to clarify this.

L141 Is inter-specific heterozygosity based on 50 SNPs with $F_{ST} > 0.6$? Please clarify

L150- The 50 SNPs... that are also consistent...- do you mean "which are also consistent" or is there a subset of 50?

L179- how did you define vicinity?

L190- how many sites are in iSTOW? Please explain what a site is. There are more than 19 points on most maps- what are they if not sites?

L191 If you replace "spatial autocorrelation" with "geographic distance" you can remove the repetitive sentence in 191-193.

L195 to add context? This aim is unclear, please explain what you mean (ideally, link to your

hypotheses).

L212 Please explain here why Chr5 differentiation block ancestry was used

L214 replace “describe” with “summarize”?

L218 Two climate predictors in one analysis or one at a time?

L223 Do you mean “consistently with published results” or you are presenting previous results? By “The mtDNA haplotype clusters”- do you mean ND2 or whole mitogenomes? Was Fig.1c based on old ND2 data (L649) or included new sequences?

L225 “Among these substitutions”- none were mentioned, which data do you mean?

L224 citation format

L236 Haida Gwaii did not differ from SOCC on EV1, correct?

L254 similar in which way? Allele frequencies?

L270 Need to mention the lack of the peak in Valdez vs iSTOW (Fig. S1D). Same is for Chr5 peak in L275.

L283 Where in the gene was the SNP? Intron or exon, amino acid changing or not?

L286-288 repeating L275-278, consolidate

L295 could you summarize here how PC1 characterizes the climate (below you explain that PC2 explains aridity)?

L312 List the three regions here please.

L320 Please provide support for this inference.

L328-329 enriched instead of concentrated? Please provide support for this statement.

L329 add “acid” after “amino”

L343 need to comment on presence of mismatched and recombinant types

L345 But the association was with PC1, whose pattern was akin to the phenotypic variation, not mitochondrial variation (Fig 5A)? This paragraph is a bit confusing

L350 Selection does not need to be invoked to explain prevalence of SOCC-like haplotypes in areas occupied by SOCC

L361 what kind of variation do you mean by “coastal habitats”?

L388-394 this paragraph is a distraction

L402 I would not consider 0.8% divergence as having “many” differences- perhaps stick to numbers?

L430 what do you mean by nuclear and genomic?

L432 What does “these” refers to?

L436 This seems like a stretch to me- I am not convinced that mitonuclear genotypes are adapted to climate at all, given how widespread the hybrid zone is.

L444 as above, I do not see any strong evidence for mitonuclear adaptation and environmental selection. Perhaps you could formulate a hypothesis for future study that would explain the observed patterns of variation.

Fig.1: Please cite the source of inferred history in the legend. Given that Fig. 1B is also replicated on Fig. 4D, it may make more sense to colour the dots on the Fig. 1B map according to the colours in the legend of Fig.1C and D- this would make geographic context easier to understand? But I see that mitochondrial frequencies are also important background. L648- which NADH gene? L653- are black circles mutation when there are >1 mutations or unsampled intermediate haplotypes? L656- Populations- how defined? same as area in L649? L657- “EV1 represents admixture ...” in which way? L660- other coastal STOW is abbreviated as oc-STOW on figures, not ocostal STOW.

Fig.2: Please state in the title how many loci this figure was based on.

Fig. 4D: Please explain what dots on the map are and what colours represent. Is top map on Fig 4D is the

same as the map on Fig. 1B? Perhaps say so

Fig. 5 Consider rephrasing the title. Fig. 5D why there are fewer turquoise dots compared to C (they are also of different colour)?

Fig. S1 L719- Valdez is listed as if it has a peak, but it does not. Please change "ocostal" to oc-STOW.
L722 add "A" after "blue boxes"

Fig. S3 explain what OC STOW is. Higher differentiation is expected in populations with lower genetic diversity (due to higher drift)

Fig. S4 title of the legend "nDNA ancestry" is misleading

Fig. S5 Change the legend to be consistent with the rest of the figures

Fig. S6- Temperature difference between what and what?

Table S1- which positions in the gene the amino acids are and what are they in each lineage?

Table S2- Please explain the source of information in the column "Related mitochondrial function"

Reviewer #3 (Remarks to the Author):

In this paper, the authors seek to assess evidence for mitonuclear coevolution in several admixed populations of warblers in the Pacific Northwest. They use F_{st} scans to identify differentiated regions of the genome between different population pairs, construct a haplotype network to assess spatial variation mitochondrial haplotypes across the region, and use a GWAS approach to look for associations between regions of the nuclear genome and mitochondrial clade. They find a region of chromosome 5 strongly associated with mitochondrial haplotype, and suggest this region may harbor genes co-adapted with mitochondrial function. They further find a correlation between mitonuclear genotype and climate, which they suggest may reflect climate adaptation.

There are some interesting ideas and results in this paper, the system is interesting and appropriate, and the analyses are mostly well done although lacking some detail. The figures are complex but visually appealing. However, many of the main points are challenging to identify amidst a lot of overly complex information and a confusing organizational structure. There are also some additional analyses that could be done to help strengthen the authors' claims of mitonuclear coevolution. Finally, throughout the paper, it is not clear to me why these results are exceptionally novel or exciting. Mitonuclear coevolution associated with climate variation has been shown more convincingly in other systems (e.g. Morales et al 2018), and small outlier regions associated with plumage differences have now been identified in many different groups of birds (e.g. Toews et al 2016, Campagna et al 2017, the authors' own previous work). I encourage the authors to make the big, novel contribution of this paper much clearer.

Below I identify some broader suggestions for improvement as well as more specific comments.

The Introduction starts with a focus on narrow hybrid zones- but this paper is not about narrow hybrid zones. The Introduction switches focus in the third paragraph to broadly admixed populations, and we never return to narrow hybrid zones elsewhere in the paper except in some brief asides. I found this

setup to be confusing- why do we need to know about narrow hybrid zones? What can we learn about species barriers and adaptation by studying these broadly admixed populations?

The Introduction also sets this system up as one with “ancient and ongoing hybridization”- but as far as I can tell there is no ongoing hybridization in the populations studied here. I think the ongoing hybridization refers to narrow hybrid zones in the southern part of the range that have been the focus of other studies, but these hybrid zones are not mentioned elsewhere in the paper except as asides. I might be misunderstanding the overall geography here though- it’s not clear to me from Figure 1 if the inland and coastal STOW populations meet and interbreed extensively, or if they are separated by the Coast Mountains. Panel 1A suggests a region of extensive sympatry and admixture, whereas the sampling in panel 1B and the text suggest the inland and coastal populations are allopatric. It would help to clarify the current geographic context and extent of mixing between the coastal and inland populations.

Triangle plot analysis: Assigning individuals to hybrid classes based on ancestry and heterozygosity will underestimate the frequency of early generation hybrids when using non-diagnostic loci. The authors cannot help that there are few differentiated loci between these populations; however, they should note that the age of hybridization based on the triangle plot should be interpreted with caution given the use of a small number of loci with $F_{st} > 0.6$ (and, as far as I can tell, no loci with fixed differences). More broadly, the authors refer throughout the ms to “ancient hybridization.” It is not clear to me what “ancient” means- it’s possible that the hybridization observed is only a few tens or hundreds of generations (a short time for a species with a one-year generation time). I think a clearer definition of “ancient” is therefore warranted. The authors could use haplotype-based approaches to get a better sense of the age of admixture in these populations, but these can be challenging with GBS data.

Mitochondrial correlations: I’m not sure I understand how the mitochondrial ancestry associations were calculated (section starting line 183 and y-axis of figure 5D). The analysis calculated mean mtDNA and nDNA ancestry in a population by averaging locus-specific ancestry among individuals, but how was locus-specific nDNA ancestry determined? Did this analysis use only the 50 differentiated SNPs from the triangle plot? Was a separate analysis done using all the loci to determine individual ancestry proportions? This section would benefit from clarification of methods.

I’m also not sure about the approach to control for geography in this section. Controlling for geography using a distance matrix and a partial Mantel test makes sense for populations in which isolation-by-distance is expected to be strong- e.g., when populations are distributed continuously across space or not separated by major barriers to gene flow. However, this approach is less effective when some populations are separated by clear geographic barriers, as is suggested for Haida Gwaii and Valdez. In such populations, we might expect to find strong differentiation across a comparatively small geographic distance due to barriers to gene flow. I think it would be useful to re-run this analysis excluding the two clearly isolated populations to see if the pattern still holds.

Regions under selection: The analysis of mitochondrial correlations identifies a region of chromosome 5 with elevated F_{st} between two of the three coastal STOW populations and SOCC. This region is significantly associated with mitochondrial haplotype. The next section looks at associations between

climate and mitonuclear genotype on chromosome 5 and suggests this region may be under selection. This analysis would be more compelling if it included π and D_{xy} , and perhaps an analysis of selective sweeps and decay of linkage disequilibrium. These analyses would bolster the argument that there is ongoing selection on mt-nDNA associations despite ongoing or past gene flow.

An additional note about the mitonuclear correlations: it looks like the F_{st} peak on chromosome 5 is absent in the Valdez-iSTOW comparison (Figure 3 & Figure 4b), but this is not mentioned anywhere in the paper. I think this needs to be addressed.

Climate analysis: The climate analysis is interesting but needs a few more details. What years were climate data extracted for? What is the resolution of the climate data? Is the resolution for the climate data at a scale relevant to the sampling and the biology of the species?

I also think it's interesting that, given the emphasis on "ancient" hybridization in these admixed populations, the authors do not look at a wider temporal range of climate variables. The correlation they find between mitonuclear genotype and present-day climate is a bit weak. Might that correlation be stronger if the authors looked at a wider range of climatic conditions that have occurred since glacial retreat?

Specific comments:

Line 150: I'm not sure what this sentence means. The triangle plot analysis was done using 50 SNPs with $F_{st} > 0.6$ between SOCC and inland STOW, with the goal of identifying hybrid classes for coastal STOW (e.g. recent F1 vs. later generation hybrids/ backcrosses). The sentence on line 150 refers to 50 SNPs from F_{st} peaks between coastal and inland STOW. Were the 50 SNPs that were differentiated between SOCC and inland STOW also differentiated between coastal and inland STOW?

Line 156- this paragraph reads like it should be in the Introduction

Lines 329-335: this section reads like Results but is in the Discussion. There is also no section of the results covering the mtDNA sequence analysis, so that section of the analysis feels rather tangential to the rest of the paper.

Lines 365-372: I don't understand this section- it seems to suggest that the Haida Gwaii and Valdez populations have a higher frequency of the SOCC mt-nDNA combination, but I don't think this is the case (or if it is it's not clear from the text). Valdez seems to have a low frequency of the SOCC combination and no outlier region on chromosome 5. This section also made me wonder if the high frequency of SOCC mtDNA on Haida Gwaii could just be due to drift/ leftover from a time when the area was inhabited by refugial SOCC. I think this supports my suggestion above to re-run the mitonuclear associations and climate analysis without the two isolated populations (Haida Gwaii and Valdez) to see if the patterns still hold.

Section starting on line 375- these two paragraphs read a bit like they came from a different paper. More generally, while I think the ASIP-RALY analysis is interesting, it is not well integrated into the paper. There isn't really enough background information for those not familiar with the system to

understand the significance of the plumage analysis, and these brief sections therefore distract from the main focus on mitonuclear coevolution.

Line 405: what is the evidence for this claim that enough time has passed for alternative haplotypes to be lost in the case of ILS?

Line 460: the idea that these populations provide natural replicates for testing selection is interesting, but the authors don't really test selection anywhere in the paper. This statement also ignores that there does not seem to be a differentiated region on chromosome 5 for one of these three populations (at least as far as I can tell from my reading), as noted above.

Dear editor and reviewers,

Thank you for the helpful comments. We have thoroughly updated the manuscript in response to the thoughtful comments of the three reviewers. Please see our response to reviewer comments in point form denoted by ">>>".

REVIEWER COMMENTS

Reviewer #1 (Remarks to the Author):

I found this to be a very important study. Along with a few other recently published papers on introgression of mt genes and mitonuclear coadaptation, this study holds the potential to reshape how evolutionary biologists think about the process of speciation.

The authors present results from genetic and statistical analyses on the nuclear and mitochondrial ancestry of two warbler species and subpopulations within those species. They find compelling evidence for ancient hybridization between ancestral populations of each species, resulting in intermediate populations with mixed ancestry. Scans of the nuclear genome revealed islands of differentiation among species and subpopulations that are associated with genes known to encode products important for mitochondrial energy metabolism. Thus, the authors suggest mitonuclear effects might have played important roles in shaping current genetic structure and plumage differences among these North American Warblers. The study seems thorough and the manuscript is well-written. I enjoyed reading the paper and I think the study will have broad implications toward understanding speciation.

>>> We are grateful for these comments from the reviewer.

I do have some suggestions for improving the paper. The bulk of these rest in interpretation and use of PCA components for correlational analyses on selection.

Major concerns:

Lines 213-219: The use of PCAs for further correlations is tricky and it is important to make expressly clear what the PC axes used for correlations represent. PCAs are most effective when used to cluster groups based on many variables in order to produce a "caricature" that is an amalgam of the variables included in the PCA. For example, a PCA of height, weight, and BMI can produce a PC1 that might represent how "fat" or how "muscular" an individual is, depending on the significance and sign (+/-) of the PC loadings.

>>> Thank you for the comment. We have now included explanations of what each of PC1 and PC2 represent. In addition, we have included statistical test of climatic PC1 and PC2 difference among parental and hybrid populations to quantitatively demonstrate climatic difference in these habitats. Please see updates in method (line 254-257), results (line 370-380), and discussion (line 423-427).

What did PC1 and PC2 represent in the PCA performed here? (ie., Does PC1 represent how hot/dry a climate is? Does PC2 represent how cold/humid a location is?) If the PC loadings are too complicated to interpret this way, then it may be a sign that biological

interpretation is lost, or that too many variables are included in the PCA to allow for biological inference from further correlations. This is why using PCAs in this way has fallen out of favor with some biologists (see Connor 2007, Chong 2018, and Hunt 2007 for more details, explanations and suggestions). I've studied table S4, and I'm not sure it is possible to get an accurate picture of what the climate PCA actually represents about climate based on those loadings. I suggest that the authors explain to the reader more clearly what is represented by PC1 and PC2. I realize this is a big ask at this stage, but I suggest that the authors consider reducing the number of climate variables to a smaller group likely to affect energetic adaptation (such as temperature based on results like Healy 2019), or to key variables meant to specifically encompass key differences between coastal and inland habitat (forestation, temperature, humidity). Narrowing the focus beforehand based on biological inference may inform interpretation of results after the fact. Alternatively, the authors might better justify the way they employed PCA.

>>> Thank you, this is a helpful comment. PC1 is largely reflecting temperature difference, while PC2 is associated moisture difference, which are important habitat climatic variation among SOCC, coastal versus inland STOW that explains the divergent selection on the mitonuclear genes. We have updated results with (line 370-380), along with Fig. 5, Fig. S6, and Table S5.

Lines 297-299: The breakdown of PC2 comes close to giving us a picture of what this PC represents. What were the signs of TD, CMI, PPT_wt, and MAP? If there were all positive, does this make PC2 represent how humid, warm, rainy and wet during the winter a location is? If this is correct then a better explanation here goes a long way toward addressing my concern in Lines 213-219 above.

>>> PC1 is negatively reflecting various temperature indices, and PC1, while PC2 is positively associated with various moisture indices. For clarity, we have included explanations in Table S5 and results. The PCA is now complimented with empirical interpretation. Please see last paragraph of the Results and the last paragraph of 'Coadaptation of mtDNA and nDNA' section of the Discussions.

Line 302-303: "Overall, coastal STOW habitat is moister and more stable in temperature" According to the PC loadings, or qualitative assessment of the climate outside of stats? Very clear descriptions would be helpful here.

>>> We have now included quantitative assessment. Please see results (Line 370-380) and Figure S6.

Minor comments:

Line 46: would "isolated" or "allopatric" be better here?

>>> Edited

Line 53: North American*

>>> added "n"

Lines 73-76: I think I understand what is being suggested here, but the language could be clearer. The assumption is that inland STOW and SOCC mtDNA haplotypes exist in an unbalanced frequency in coastal STOW based on location. This variation in frequency from location to location might spur differentiation WITHIN the coastal STOW

subpopulations - presumably based on local adaptation? Is this the current hypothesis?

>>> Thank you for pointing this out. We have rewritten this sentence for clarity.

Line 82-82: For goal 3, is this within just coastal STOW? or across all groups in this study?

I'm not sure this becomes clearer after reading the discussion either

>>> Across all groups. We have modified this last paragraph of introduction.

Line 96: What tissue was used for these samples?

>>> added "muscle"

Line 102: Any noticeable differences among old frozen samples and new blood samples?

>>> We have checked old versus new samples from the same geographical range, there was no observable difference by tissue types. We have included the information in the last sentence of this paragraph.

line 126: examined*

>>>added "d"

Line 195: nDNA*

>>> edited.

In methods: It is a matter of personal preference, but I argue that software (including R packages) should be cited in manuscripts. Software racks up thousands of citations, but some people utilize functions in academic search engines to see what kind of studies software is currently being used for. This not only helps other researchers decide whether or not to use a package, but it also can increase traffic to studies that cite software.

>>> Indeed, we have now included all the software citations.

Line 263: melanin pigmentation? Carotenoid pigmentation? STOW and SOCC differ in both.

>>> Both, we have included the information.

Literature cited in this review

Conner, J. K. (2007). A tale of two methods: putting biology before statistics in the study of phenotypic evolution. *Journal of Evolutionary Biology*, 20(1), 17-19.

Chong, V. K., Fung, H. F., & Stinchcombe, J. R. (2018). A note on measuring natural selection on principal component scores. *Evolution letters*, 2(4), 272-280.

Harada, A. E., Healy, T. M., & Burton, R. S. (2019). Variation in thermal tolerance and its relationship to mitochondrial function across populations of *Tigriopus californicus*. *Frontiers in physiology*, 10, 213.

Hunt, J., Wolf, J. B., & Moore, A. J. (2007). The biology of multivariate evolution. *Journal of evolutionary biology*, 20(1), 24-27.

Reviewer #2 (Remarks to the Author):

The paper "Signatures of mitonuclear coevolution in a warbler species complex" explores a hot topic in evolutionary biology: coevolved mitonuclear genotypes as a mechanism of climate adaptation. This is done in a system, where hybridization between two bird species that are phenotypically distinct but genetically similar resulted in divergent mitochondrial

haplotypes co-existing across the broad coastal hybrid zone. Authors tested whether some nuclear genes show differentiation according to the mitochondrial haplotypes, as would be expected under selection for co-evolved mito-nuclear allele combinations to maintain proper mitochondrial function. They found a small genomic region on Chromosome 5, enriched with genes associated with mitochondrial function, with alleles strongly correlated with mtDNA haplotype. Variation at one nDNA SNP in that region had a near-perfect correlation with mtDNA haplotype; this SNP was located in PAPLN gene with function in energy-related signalling transduction, including in mitochondria. Together with some evidence for correlation between climatic and mtDNA variation, these results are used to conclude that adaptation to environment drives mitonuclear differentiation and introgression.

The evidence for selection for co-function of mtDNA and nuclear gene block on Chr5 appears strong, with only few mismatches between mtDNA and PAPLN SNP observed in the data (Fig 4). But the evidence for environmental selection for different mitochondrial function (i.e. better fit of the SOCC-like mitonuclear genotype to coastal climate; L361) is not so convincing. If hybridization is ancient (e.g. L13, 414), and climatic selection on a mitonuclear genotype is current, then why selection did not remove the less fit mitonuclear genotype?

>>> There is a climatic gradient from coastal to inland habitat where STOW reside. Despite the divergent climate-associated variation (Fig. S6), it is unlikely that selection could completely remove incompatible mitonuclear genotypes. Despite that random mating could break down the mitonuclear associations at every generation, we still observed significant mitonuclear association, suggesting that the selection is very strong, which is consistent with the clear signature of divergent selection and selective sweep within the chr5 gene block (Fig. 5), as well as the significant partial mantel association between mitonuclear ancestry and climatic variation across space (Results, line 370-383). There is a possibility of stabilizing selection within the coastal STOW range, which is climatically differentiated from either of the parental range. We have included more discussions (Line 448-459) around this interest.

Significant correlation between climate PC1 and mitonuclear genotype appears to be driven by similarity between iSTOW and cSTOW, whereas climate PC2, for which cSTOW and SOCC were similar, was not significantly associated with mitonuclear ancestry. This does not support the better fit of the SOCC-like mitonuclear genotype to coastal environments.

>>> Indeed, c-STOW habitat is moister than i-STOW habitat and colder than the SOCC habitat, while the site level mitonuclear ancestry of i-STOW, c-STOW, and i-STOW are different. However, despite that the c-STOW is intermediate for mitonuclear ancestry, the climate PC1 or PC2 of c-STOW habitat is not intermediate. While it is impossible to completely isolate the relative climatic divergence among SOCC, c-STOW, and i-STOW habitat, it should not be isolated because it could underpin the selection that have shaped divergent mitonuclear coadaptation in these populations. With that said, we have controlled for spatial autocorrelation in the climate-ancestry association test to control for the confounding variable of geographical distance (method line 260-263). In addition, we have included discussion (line 423-454) to provide more context and synthesis of what the climate-mitonuclear ancestry would imply.

Is it possible that selection against less fit iSTOW-like mitonuclear genotype was historically strong but is currently weak due to post-LGM changes in climate?

>>> The current mitonuclear ancestry should respond to the current (or recent past) climatic condition. The post-LGM mitonuclear ancestry could be very different from the current estimate and should be evaluate in the historical context. We have included the comment in discussion (Line 435-447).

Are estimates of gene flow between other iSTOW and cSTOW available to support or refute the idea that ongoing gene flow supplements coastal populations with iSTOW alleles (L364)? Because Valdez and Haida Gwaii, which are said to be more sheltered from current gene flow from iSTOW, also have both mtDNA haplotypes, selection for SOCC-like haplotypes in coastal environments does not appear likely. Thus, I suggest to tone down inferences of climate-related selection and adaptation.

>>> We included genomic cline analyses of chr5 gene block and mtDNA (Fig. 5), which demonstrate significantly negative alpha parameter, consistent with asymmetrical introgression of the SOCC mitonuclear combination into the coastal STOW relative to ancestry background. The association of mitonuclear ancestry and climatic variation (after controlling for spatial autocorrelation) indicates that the asymmetrical introgression might be climate-facilitated, although more finetuned future sampling and experiment would be needed to test this idea. Therefore, we have to be careful whenever climate association is mentioned to avoid overinterpretation, as “climate-associated mitonuclear ancestry”, so that we present the result mitonuclear climate association as it is, while discuss possible predictions for future study. We have also modified discussion (Line 448-459) to account for anomalies and alternative predictions (please see sentences including line 395-398 and line 444-459, in which we accounted for alternatives and future directions).

It was surprising that such low mitochondrial divergence (0.8%, 5 fixed amino acid substitutions) can potentially result in strong mitonuclear incompatibilities- is there any support for the mitochondrial haplotypes (and potentially respective mitonuclear genes) being functionally different, e.g. based on physicochemical properties (TREESAAP, Woolley et al. 2003) or functional analyses (PROVEAN, Choi et al. 2012)?

>>> Thank you for the suggestion, we have included both PROVEAN and TREESAAP analyses, the ND2 substitutions were non-neutral. We have added method line 237-246, results line 275-283, and Table S2, S6, S7.

Valdez population did not have a Chr5 peak when compared to iSTOW (Fig. 3, S1)- why could this be? Genomic FST peaks could arise due to low level of variation in the region- does FST peak on Chr5 also correspond to a peak in average sequence divergence in this region, as would be expected under divergent selection for different mitonuclear combinations? Or, could the FST peak on Chr5 (and its correlation with mtDNA) result from sex-linkage?

>>> The genome-wide differentiation is quite high between Valdez and inland STOW (higher than any other comparisons to inland STOW). We have included explanations why this population could be geographically isolated by coastal mountains. Although this region did not come out as a FST peak within Valdez (which has almost even fractions of SOCC and inland STOW mtDNA ancestry), the mitonuclear association within Valdez is very strong (Fig. 4 C). In

addition, we further validated signature of selection with nucleotide diversity scan and LD map (Fig.5, S4) within Valdez. There is clear signature of selective sweep in this chr5 gene block within Valdez. We further included genomic cline analysis (Fig. 5, Table S3), which also supports divergent selection on this gene region.

My other comments are minor, mainly on clarity of expression and better transparency of methods and results. The paper is well written, methodologically sound and is a pleasure to read. This work will make a significant contribution to mito-nuclear coevolution and speciation research.

>>> Thank you for the insightful evaluation.

Other comments:

L10-12 please split this sentence. You probably don't have to abbreviate inland and coastal STOW in the abstract.

>>> changed.

L14 Perhaps start this sentence with "One of the few highly-differentiated nDNA regions...". Also, please rephrase to clarify what covaries with what (presumably, variation in nDNA with variation in mtDNA). Same comment for L20, L316: a region cannot correlate with mtDNA haplotypes, variation within a region can.

>>> Edited.

L29-32 Split this sentence. In L31 specify that you mean the mitochondrial and nuclear genomes.

>>> specified.

L49 do you mean selection AGAINST suboptimal combinations of ALLELES?

>>> indeed, changed.

L54 add "coastal" after "inhabits"?

>>> added

L58 move (abbreviated as inland STOW)" to L56 after "inland population".

>>> done.

L70 add "genomic" before "regions" (unless you mean geographic?)

>>> done.

L71 ref 30 is a bioRxiv version of 29- is this an error? Same in L256

>>> yes, deleted.

L77-83 The section describing author's motivation for the study needs strengthening. Why would incompatibilities be expected in a stable hybrid zone, where mtDNA haplotypes are only 0.8% different? Can Question 2 be answered with current correlative approaches? It might be better to formulate hypotheses about drivers of the observed patterns, and seek support for these hypotheses. It could be also helpful to briefly summarize the approaches taken to address these questions in the end of the Introduction.

>>> Thank you for the comment. Revised drastically please see the last paragraph of introduction.

L134 It was unclear what "population" refers to here- are populations based on nuclear genetics or geography? It was also not clear what you call sites and how many of them are plotted on the maps- it would be good to have a table somewhere in Supp Mat to clarify this.

>>> Thanks, we have revised this sentence for clarity. Fig. S3 summarizes the pairwise differentiation we examined.

L141 Is inter-specific heterozygosity based on 50 SNPs with $F_{ST} > 0.6$? Please clarify

>>> Included clarification, thanks!

L150- The 50 SNPs... that are also consistent...- do you mean “which are also consistent” or is there a subset of 50?

>>> Clarified, please see line 150-153.

L179- how did you define vicinity?

>>> We have rewritten for clarity.

L190- how many sites are in iSTOW? Please explain what a site is. There are more than 19 points on most maps- what are they if not sites?

>>> This is because to run mitonuclear site association, we have to rely on sites of which we have estimates of both mtDNA and nDNA ancestry. Taking the intersect, there were 19 sites remaining. We have modified line 191-194 for clarification.

L191 If you replace “spatial autocorrelation” with “geographic distance” you can remove the repetitive sentence in 191-193.

>>> replace and removed

L195 to add context? This aim is unclear, please explain what you mean (ideally, link to your hypotheses).

>>> edited

L212 Please explain here why Chr5 differentiation block ancestry was used

>>> Added explanation. Please see Line 181-184.

L214 replace “describe” with “summarize”?

>>> This part was edited, so this comment is no longer relevant.

L218 Two climate predictors in one analysis or one at a time?

>>> We have changed climate analysis and this comment is no longer relevant.

L223 Do you mean “consistently with published results” or you are presenting previous results?

>>> Yes, added.

By “The mtDNA haplotype clusters”- do you mean ND2 or whole mitogenomes? Was Fig.1c based on old ND2 data (L649) or included new sequences?

>>> Yes, ND2, we now have added specification.

L225 “Among these substitutions”- none were mentioned, which data do you mean?

>>> Added specification.

L224 citation format

>>> changed.

L236 Haida Gwaii did not differ from SOCC on EV1, correct?

>>> Yes, we have deleted this sentence as it is confusing and unnecessary.

L254 similar in which way? Allele frequencies?

>>> Rephrased, thanks.

L270 Need to mention the lack of the peak in Valdez vs iSTOW (Fig. S1D). Same is for Chr5 peak in L275.

>>> Included.

L283 Where in the gene was the SNP? Intron or exon, amino acid changing or not?

>>> We have included such information in Table S3 along with the gene list.

L286-288 repeating L275-278, consolidate

>>> done.

L295 could you summarize here how PC1 characterizes the climate (below you explain that PC2 explains aridity)?

>>> We have added PC characterization. Results Line 370-380.

L312 List the three regions here please.

>>> listed.

L320 Please provide support for this inference.

>>> Added Fig. 5.

L328-329 enriched instead of concentrated? Please provide support for this statement.

>>> Added "17/22".

L329 add "acid" after "amino"

>>> The sentence was deleted, thus this is no longer applicable.

L343 need to comment on presence of mismatched and recombinant types

>>> Added.

L345 But the association was with PC1, whose pattern was akin to the phenotypic variation, not mito-nuclear variation (Fig 5A)? This paragraph is a bit confusing

>>> The mitonuclear ancestry was correlated with climate PC1. We included "(Fig. 5F)" and have rewritten the sentence for clarity.

L350 Selection does not need to be invoked to explain prevalence of SOCC-like haplotypes in areas occupied by SOCC

>>> We have now included explicit tests of selection with pi, LD, and genomic cline.

L361 what kind of variation do you mean by "coastal habitats"?

>>> changed to "climatic conditions"

L388-394 this paragraph is a distraction

>>> We have deleted this paragraph.

L402 I would not consider 0.8% divergence as having "many" differences- perhaps stick to numbers?

>>> Edited.

L430 what do you mean by nuclear and genomic?

>>> mitochondrial and nuclear, edited.

L432 What does "these" refers to?

>>> cited Fig. 1D.

L436 This seems like a stretch to me- I am not convinced that mitonuclear genotypes are adapted to climate at all, given how widespread the hybrid zone is.

>>> This is a prediction for future testing. We found significant correlation between mitonuclear ancestry and breeding climatic condition across sites, pointing to the possibility of mitonuclear climate-associated adaptation (as discussed in line. 423-447). We have modified this section in discussion to be clear that the correlational results generate a climate adaptation prediction for future test. In addition, we have included discussion about alternatives, including potential balancing selection in the coastal STOW that is different from both inland STOW and SOCC (line 448-459).

L444 as above, I do not see any strong evidence for mitonuclear adaptation and environmental selection. Perhaps you could formulate a hypothesis for future study that would explain the observed patterns of variation.

>>> Thank you, we have toned down implication about the climate-association in the discussion and included more future direction suggestions (line 423-459). In addition, we have included

various selection tests in this revision on the mtDNA (Table S2) and chr5 candidate gene block (Fig. 5, S4), where we found clear signatures of selection in mitonuclear genes. We have rewritten this section to state and synthesize the evidence of mitonuclear co-adaption.

Fig.1: Please cite the source of inferred history in the legend.

>>>Cited.

Given that Fig. 1B is also replicated on Fig. 4D, it may make more sense to colour the dots on the Fig. 1B map according to the colours in the legend of Fig.1C and D- this would make geographic context easier to understand? But I see that mitochondrial frequencies are also important background.

>>> We choose to use discrete color for effect

L648- which NADH gene?

>>> ND2, edited.

L653- are black circles mutation when there are >1 mutations or unsampled intermediate haplotypes?

>>> when there are >1 mutations, there would be >1 block dots. Unsampled intermediates is not relevant for one mitochondrial gene.

L656- Populations- how defined? same as area in L649?

>>> edited.

L657- “EV1 represents admixture ...” in which way?

>>> Edited.

L660- other coastal STOW is abbreviated as oc-STOW on figures, not ocostal STOW.

>>> Edited.

Fig.2: Please state in the title how many loci this figure was based on.

>>> Added

Fig. 4D: Please explain what dots on the map are and what colours represent. Is top map on Fig 4D is the same as the map on Fig. 1B? Perhaps say so

>>> Added

Fig. 5 Consider rephrasing the title. Fig. 5D why there are fewer turquoise dots compared to C (they are also of different colour)?

>>> Rephrased. C plotted individual ancestry, while D plotted site mean ancestry. We have edited Fig. 5 caption to clarify this point.

Fig. S1 L719- Valdez is listed as if it has a peak, but it does not. Please change “ocostal” to oc-STOW. L722 add “A” after “blue boxes”

>>> Removed Valdez. Added, thanks!

Fig. S3 explain what OC STOW is. Higher differentiation is expected in populations with lower genetic diversity (due to higher drift)

>>> Explained. Included the drift explanation.

Fig. S4 title of the legend “nDNA ancestry” is misleading

>>> Indeed, updated.

Fig. S5 Change the legend to be consistent with the rest of the figures

>>> changed, thanks.

Fig. S6- Temperature difference between what and what?

>>> this is no longer relevant as the climate analysis is updated.

Table S1- which positions in the gene the amino acids are and what are they in each lineage?

>>> We have included table S2 for such information.

Table S2- Please explain the source of information in the column "Related mitochondrial function"

>>>Included, thanks.

Reviewer #3 (Remarks to the Author):

In this paper, the authors seek to assess evidence for mitonuclear coevolution in several admixed populations of warblers in the Pacific Northwest. They use Fst scans to identify differentiated regions of the genome between different population pairs, construct a haplotype network to assess spatial variation mitochondrial haplotypes across the region, and use a GWAS approach to look for associations between regions of the nuclear genome and mitochondrial clade. They find a region of chromosome 5 strongly associated with mitochondrial haplotype, and suggest this region may harbor genes co-adapted with mitochondrial function. They further find a correlation between mitonuclear genotype and climate, which they suggest may reflect climate adaptation.

There are some interesting ideas and results in this paper, the system is interesting and appropriate, and the analyses are mostly well done although lacking some detail. The figures are complex but visually appealing.

However, many of the main points are challenging to identify amidst a lot of overly complex information and a confusing organizational structure.

>>> Thanks for pointing this out. We have substantially edited the main text to account for this concern.

There are also some additional analyses that could be done to help strengthen the authors' claims of mitonuclear coevolution.

>>> Thank you for the helpful suggestions. We have included the new analyses suggested (see below for specifics). Primarily, we have included explicit selection tests both in terms of the mitochondrial variation and the chr5 variation.

Finally, throughout the paper, it is not clear to me why these results are exceptionally novel or exciting. Mitonuclear coevolution associated with climate variation has been shown more convincingly in other systems (e.g. Morales et al 2018), and small outlier regions associated with plumage differences have now been identified in many different groups of birds (e.g. Toews et al 2016, Campagna et al 2017, the authors' own previous work). I encourage the authors to make the big, novel contribution of this paper much clearer.

>>> The divergent climate-associated mitonuclear coevolution underlying cryptic differentiation between natural hybrid and parental vertebrate populations is quite novel. We have revised the manuscript substantially to highlight the knowledge gap that this study fulfills. Please see abstract, line 28-31 in introduction, conclusion line 528-543.

Below I identify some broader suggestions for improvement as well as more specific comments.

The Introduction starts with a focus on narrow hybrid zones- but this paper is not about narrow hybrid zones. The Introduction switches focus in the third paragraph to broadly admixed populations, and we never return to narrow hybrid zones elsewhere in the paper except in some brief asides. I found this setup to be confusing- why do we need to know about narrow hybrid zones?

>>> Indeed, we have deleted the part about narrow hybrid zones.

What can we learn about species barriers and adaptation by studying these broadly admixed populations?

>>> That is a good point, we have added more background for this. Please see line 41-49.

The Introduction also sets this system up as one with “ancient and ongoing hybridization”- but as far as I can tell there is no ongoing hybridization in the populations studied here. I think the ongoing hybridization refers to narrow hybrid zones in the southern part of the range that have been the focus of other studies, but these hybrid zones are not mentioned elsewhere in the paper except as asides.

>>> Indeed, we have now removed “ongoing”.

I might be misunderstanding the overall geography here though- it’s not clear to me from Figure 1 if the inland and coastal STOW populations meet and interbreed extensively, or if they are separated by the Coast Mountains. Panel 1A suggests a region of extensive sympatry and admixture, whereas the sampling in panel 1B and the text suggest the inland and coastal populations are allopatric. It would help to clarify the current geographic context and extent of mixing between the coastal and inland populations.

>>> The coastal and inland STOW are mostly not geographically isolated, except the Valdez (surrounded by coastal mountains) and Haida Gwaii (surrounded by the ocean) populations. They were called one species because of the identical appearances. However, the genetic analysis suggest that they are not. There could still be gene flow between coastal and inland populations, but the Fig. 1B sampling was designed to effectively represent the extreme coastal versus inland of STOW range. Future studies could investigate the populations between the coastal versus inland extremes. We have included more clarifications in line 62-65 and Fig. 1 caption (line 758-759).

Triangle plot analysis: Assigning individuals to hybrid classes based on ancestry and heterozygosity will underestimate the frequency of early generation hybrids when using non-diagnostic loci. The authors cannot help that there are few differentiated loci between these populations; however, they should note that the age of hybridization based on the triangle plot should be interpreted with caution given the use of a small number of loci with $F_{st} > 0.6$ (and, as far as I can tell, no loci with fixed differences). More broadly, the authors refer throughout the ms to “ancient hybridization.” It is not clear to me what “ancient” means- it’s possible that the hybridization observed is only a few tens or hundreds of generations (a short time for a species with a one-year generation time). I think a clearer definition of “ancient” is therefore warranted. The authors could use haplotype-based approaches to get a better sense of the age of admixture in these populations, but these can be challenging with GBS data.

>>> That is a good point, we have now omitted the term ancient hybrids, and refer it as hybrid in

origin instead. The mitonuclear coadaptation within admixed population is the key result of this paper, which does not depend on the age of the admixture. The admixture is likely happened 5000 years ago based on boreal forest postglacial expansion (Rohwer 2001). The low heterozygosity (relative to a value close to 1 for early generation hybrids) and intermediate admixture proportions suggest that the admixture happened a long time ago (line 145-149), which is subject to estimate for future studies. Haplotype-based admixture dating is beyond the scope of the study, but we have incorporated the heterozygosity and ancestry inference with the postglacial boreal forest invasion, along with future study incentives in the discussion (line 494-503).

Mitonuclear correlations: I'm not sure I understand how the mitonuclear ancestry associations were calculated (section starting line 183 and y-axis of figure 5D). The analysis calculated mean mtDNA and nDNA ancestry in a population by averaging locus-specific ancestry among individuals, but how was locus-specific nDNA ancestry determined?

>>> The locus-specific ancestry was determined by first differentiate the inland STOW versus SOCC alleles, and assigning the ancestry of 0 for homozygous SOCC, 1 for homozygous inland STOW and 0.5 for heterozygotes. We have rewritten this section (Line 183-186) for clarity.

Did this analysis use only the 50 differentiated SNPs from the triangle plot? Was a separate analysis done using all the loci to determine individual ancestry proportions? This section would benefit from clarification of methods.

>>>For site level mitonuclear ancestry association, we narrowed down to only use the chr5 FST peak which harbours 15 SNPs (Fig. 4C) (out of the 50 SNPs genome-wide with $F_{ST} > 0.6$). This is because from the mitonuclear GWAS, we have already found the 'candidate gene block' that is significantly covarying with mtDNA. We have rewritten this section for clarity (line 341-350) as well as caption Fig. 4.

I'm also not sure about the approach to control for geography in this section. Controlling for geography using a distance matrix and a partial Mantel test makes sense for populations in which isolation-by-distance is expected to be strong- e.g., when populations are distributed continuously across space or not separated by major barriers to gene flow. However, this approach is less effective when some populations are separated by clear geographic barriers, as is suggested for Haida Gwaii and Valdez. In such populations, we might expect to find strong differentiation across a comparatively small geographic distance due to barriers to gene flow. I think it would be useful to re-run this analysis excluding the two clearly isolated populations to see if the pattern still holds.

>>> Thank you for the helpful comments. We did rerun the analysis without the two sites and the result still holds. We have edited method (line 195-198) and results for this change (line 347-350).

Regions under selection: The analysis of mitonuclear correlations identifies a region of chromosome 5 with elevated F_{ST} between two of the three coastal STOW populations and SOCC. This region is significantly associated with mitochondrial haplotype. The next section looks at associations between climate and mitonuclear genotype on chromosome 5 and suggests this region may be under selection. This analysis would be more compelling if it included π and D_{xy} , and perhaps an analysis of selective sweeps and decay of linkage disequilibrium. These analyses would bolster the argument that there is ongoing selection

on mt-nDNA associations despite ongoing or past gene flow.

>>> Thank you for the great idea! We have included suggested analysis. Indeed, there was a reduction of π and elevation of LD within the chr5 1.2Mb candidate gene block. Please see Figure 5, method (Line 200-215), results (Line 351-368), and discussion (Line 410-412).

An additional note about the mitonuclear correlations: it looks like the F_{st} peak on chromosome 5 is absent in the Valdez-iSTOW comparison (Figure 3 & Figure 4b), but this is not mentioned anywhere in the paper. I think this needs to be addressed.

>>> We think this is because there is almost even SOCC versus STOW mt haplotype within Valdez. If you look within Valdez, there is strong mitonuclear association in this region. We now have included the explanation.

Climate analysis: The climate analysis is interesting but needs a few more details. What years were climate data extracted for? What is the resolution of the climate data? Is the resolution for the climate data at a scale relevant to the sampling and the biology of the species?

>>> Thank you for pointing this out. We have now included the details in method 248-263, results 183-186, and discussion line 423-459, Fig. S6.

I also think it's interesting that, given the emphasis on "ancient" hybridization in these admixed populations, the authors do not look at a wider temporal range of climate variables. The correlation they find between mitonuclear genotype and present-day climate is a bit weak. Might that correlation be stronger if the authors looked at a wider range of climatic conditions that have occurred since glacial retreat?

>>> This is an interesting point. Here with the contemporary climate data, we demonstrated that the contemporary mitonuclear ancestry covary with the current climate. The prehistory climate is surely different from now, but the mitonuclear ancestry could be also different. It is beyond the scope of this paper to do a full analysis of past climates and how the current population genomics map onto them. We have included more discussion to parse out the spatial and temporal variation of mitonuclear ancestry and climatic conditions (line 435-459).

Specific comments:

Line 150: I'm not sure what this sentence means. The triangle plot analysis was done using 50 SNPs with $F_{ST} > 0.6$ between SOCC and inland STOW, with the goal of identifying hybrid classes for coastal STOW (e.g. recent F1 vs. later generation hybrids/ backcrosses). The sentence on line 150 refers to 50 SNPs from F_{st} peaks between coastal and inland STOW. Were the 50 SNPs that were differentiated between SOCC and inland STOW also differentiated between coastal and inland STOW?

>>> Not necessarily. We found the most differentiated SNPs between the parental populations and examined how they segregate in coastal STOW admixture. Clarification is added to line 147-157.

Line 156- this paragraph reads like it should be in the Introduction

>>> We have moved to last paragraph of introduction and drastically revised the last paragraph.

Lines 329-335: this section reads like Results but is in the Discussion. There is also no section of the results covering the mtDNA sequence analysis, so that section of the analysis feels rather tangential to the rest of the paper.

>>> We have trimmed this paragraph and included more functional mitochondrial sequence analysis, the signature of selective sweep, mitonuclear genomic cline analysis, synthesizing the inference of mitonuclear coadaptation. Please see Fig. 5, and line of discussion line 400-422.

Lines 365-372: I don't understand this section- it seems to suggest that the Haida Gwaii and Valdez populations have a higher frequency of the SOCC mt-nDNA combination, but I don't think this is the case (or if it is it's not clear from the text). Valdez seems to have a low frequency of the SOCC combination and no outlier region on chromosome 5. This section also made me wonder if the high frequency of SOCC mtDNA on Haida Gwaii could just be due to drift/ leftover from a time when the area was inhabited by refugial SOCC. I think this supports my suggestion above to re-run the mitonuclear associations and climate analysis without the two isolated populations (Haida Gwaii and Valdez) to see if the patterns still hold.

>>> We have dropped Haida Gwaii and Valdez, the pattern still holds. Please see methods (line 195-198) and results (line 347-350).

Section starting on line 375- these two paragraphs read a bit like they came from a different paper. More generally, while I think the ASIP-RALY analysis is interesting, it is not well integrated into the paper. There isn't really enough background information for those not familiar with the system to understand the significance of the plumage analysis, and these brief sections therefore distract from the main focus on mitonuclear coevolution.

>>> We have now reduced the section on ASIP-RALY, and integrated it better into the paper. As the reviewer notes, the ASIP-RALY information is interesting, and we think many readers will be interested in the comparison of patterns of geographic variation between ASIP-RALY and the chr5 gene block.

Line 405: what is the evidence for this claim that enough time has passed for alternative haplotypes to be lost in the case of ILS?

>>> Under Wright-Fisher model, we could estimate the generation numbers until a frequent variant to be completely lost in a population, based on effective population size estimate (based in nucleotide diversity estimate and mutation rate). For the effective population size of 340136.1 for inland STOW, it would take 235309 generations to completely lose the SOCC variant. We have included the explains in line 486-489.

Line 460: the idea that these populations provide natural replicates for testing selection is interesting, but the authors don't really test selection anywhere in the paper. This statement also ignores that there does not seem to be a differentiated region on chromosome 5 for one of these three populations (at least as far as I can tell from my reading), as noted above.

>>> Thank you for the great idea. We have now included analyses of selective sweep detection, in terms of π , LD, and genomic cline analysis. Please see methods (Line 199-221), results (Line 351-368), discussion (line 410-413) and Fig. 5, S4.

REVIEWERS' COMMENTS

Reviewer #1 (Remarks to the Author):

The authors have done a commendable job of revising their manuscript in light of suggests of reviewers. I have no further suggestions for improving this paper.

I will reiterate my opinion that the findings of this study are novel and hold the potential to add substantially to our understanding of gene flow between populations of birds in particular and perhaps animals in general. There is some precedence for this study in the eastern yellow robin studies from coastal Australia, but that study involves two diverging populations, not two ancient independent lineages that have exchanged mitochondrial and nuclear genes in a specific pattern that seems to be driven by climate adaptation. It is only through the publication of the details of gene exchange in multiple animal systems that we will begin to understand the importance and ubiquity of mitonuclear gene interaction in relation to the interactions of other genes.

Reviewer #2 (Remarks to the Author):

I was Reviewer #2 of the previous submission. The authors did a great job adding new analyses and making the manuscript clearer. All my queries were adequately addressed. Addition of selection and genomic cline analyses strengthened the case for climatic adaptation. I am very happy to recommend acceptance of this fascinating and important work. Looking forward to seeing it in print!

Below are several very minor suggestions which authors can easily address:

L15 observations, not observation

L31 have, not has

L40 experiencing or residing in

L43 become? Perhaps acquire? Or 'units' instead of trajectories?

L98 replace 'Among these individuals' with 'from samples'

L334 in Valdes vs inland STOW comparison... because allele frequencies are less dissimilar?

L356 something is wrong with this sentence. Region between 2.5Mb and 7Mb?

L373 There was significant climatic PC1...? Do you mean significant difference in PC1 values?

L395 co-adaptation- spelling. We found –add 'that'

L406 ...involved in ATP and NADH activities

Suppl table S3 (excel spreadsheet) is labelled S2. By 'extron' do you mean exon?

Alexandra Pavlova

Reviewer #3 (Remarks to the Author):

The authors are to be commended on a much improved ms. The Introduction is much clearer and better articulates the relevance of studying broadly admixed populations; the ASIP-RALY results are better integrated; and the analysis of pi and LD strengthens the inference that the candidate region on chr5 is

under selection. I enjoyed reading this revision.

My only remaining substantial comment is that more detail is needed about the (new) genomic cline analysis. To my knowledge, the vast majority of applications of this approach use the `bgc` program (Gompert and Buerkle 2012), which accounts for genotype uncertainty in GBS data. The authors instead used a standard linear mixed modeling package for this analysis, so considerably more detail is needed about how the models were specified to ensure reproducibility. They should also note how many loci were assessed in the cline analysis. The genomic cline model can produce false positives under certain conditions (Gompert and Buerkle 2011, see e.g. MacFarlane et al 2021 *Molecular Ecology* for detailed discussion), so some mention of whether there may be alternative interpretations of these clines is also warranted.

Line 185: locus specific ancestry of all loci, just differentiated loci, or just loci in the chr5 block?

Line 334: a bit more context here- what does it mean that the ancestry is 0.4 in Valdez? How might that affect the absence of the chr5 peak?

Lines 417-422: the last three sentences of this paragraph are rather clunky

REVIEWERS' COMMENTS

Reviewer #1 (Remarks to the Author):

The authors have done a commendable job of revising their manuscript in light of suggests of reviewers. I have no further suggestions for improving this paper.

I will reiterate my opinion that the findings of this study are novel and hold the potential to add substantially to our understanding of gene flow between populations of birds in particular and perhaps animals in general. There is some precedence for this study in the eastern yellow robin studies from coastal Australia, but that study involves two diverging populations, not two ancient independent lineages that have exchanged mitochondrial and nuclear genes in a specific pattern that seems to be driven by climate adaptation. It is only through the publication of the details of gene exchange in multiple animal systems that we will begin to understand the importance and ubiquity of mitonuclear gene interaction in relation to the interactions of other genes.

>>> Thank you, we appreciate the big-picture insights.

Reviewer #2 (Remarks to the Author):

I was Reviewer #2 of the previous submission. The authors did a great job adding new analyses and making the manuscript clearer. All my queries were adequately addressed. Addition of selection and genomic cline analyses strengthened the case for climatic adaptation. I am very happy to recommend acceptance of this fascinating and important work. Looking forward to seeing it in print!

>>> Thank you for the helpful ideas and suggestions.

Below are several very minor suggestions which authors can easily address:

L15 observations, not observation

>>> added "s".

L31 have, not has

>>> changed.

L40 experiencing or residing in

>>> changed to "experiencing".

L43 become? Perhaps acquire? Or 'units' instead of trajectories?

>>> changed.

L98 replace 'Among these individuals' with 'from samples'

>>> replaced.

L334 in Valdez vs inland STOW comparison... because allele frequencies are less dissimilar?

>>> Because there was less SOCC mtDNA introgression in Valdez, so that the chr5 peak did not come out strongly in the Valdez-inland STOW comparison. However, this gene block significantly covaries with mtDNA ancestry within Valdez. We have modified the sentence to clarify this point.

L356 something is wrong with this sentence. Region between 2.5Mb and 7Mb?

>>> Edited.

L373 There was significant climatic PC1...? Do you mean significant difference in PC1 values?

>>> **Yes, edited.**

L395 co-adaptation- spelling. We found –add ‘that’

>>> **Thanks, changed.**

L406 ...involved in ATP and NADH activities

>>> **Edited.**

Suppl table S3 (excel spreadsheet) is labelled S2. By ‘extron’ do you mean exon?

>>> **Thanks. Edited.**

Alexandra Pavlova

Reviewer #3 (Remarks to the Author):

The authors are to be commended on a much improved ms. The Introduction is much clearer and better articulates the relevance of studying broadly admixed populations; the ASIP-RALY results are better integrated; and the analysis of pi and LD strengthens the inference that the candidate region on chr5 is under selection. I enjoyed reading this revision.

>>> **Thank you for the helpful analytical ideas and writing suggestions. We are glad with the improvements.**

My only remaining substantial comment is that more detail is needed about the (new) genomic cline analysis. To my knowledge, the vast majority of applications of this approach use the bgc program (Gompert and Buerkle 2012), which accounts for genotype uncertainty in GBS data. The authors instead used a standard linear mixed modeling package for this analysis, so considerably more detail is needed about how the models were specified to ensure reproducibility. They should also note how many loci were assessed in the cline analysis. The genomic cline model can produce false positives under certain conditions (Gompert and Buerkle 2011, see e.g. MacFarlane et al 2021 Molecular Ecology for detailed discussion), so some mention of whether there may be alternative interpretations of these clines is also warranted.

>>> **Thank you for pointing this out. We have included a paragraph with details. Please see line 520-535.**

Line 185: locus specific ancestry of all loci, just differentiated loci, or just loci in the chr5 block?

>>> **chr5 gene block. We have added clarification.**

Line 334: a bit more context here- what does it mean that the ancestry is 0.4 in Valdez? How might that affect the absence of the chr5 peak?

>>> **There was less SOCC mt introgression in Valdez, so that even though the ancestry within chr5 gene block covaries with mtDNA ancestry, this region does not come out as a peak of differentiation between Valdez and inland STOW. We have reworded the sentence with explanations.**

Lines 417-422: the last three sentences of this paragraph are rather clunky

>>> **Edited.**